# Stable carbon and nitrogen isotopic composition of leaves, litter, and soils of various ecosystems along an elevational and land-use gradient at Mount Kilimanjaro, Tanzania

Friederike Gerschlauer[1], Gustavo Saiz[2,1*], David Schellenberger Costa[3], Michael Kleyer[3], Michael Dannenmann[1], Ralf Kiese[1]

[1] Institute of Meteorology and Climate Research, Karlsruhe Institute of Technology, Garmisch-Partenkirchen, Germany

[2] Department of Environmental Chemistry, Faculty of Sciences, Universidad Católica de la Santísima Concepción, Concepción, Chile

[3] Department of Biology and Environmental Sciences, University of Oldenburg, Oldenburg, Germany

*Correspondence to:* Gustavo Saiz (gsaiz@ucsc.cl)

**Abstract**

Variations in the stable isotopic composition of carbon ($\delta^{13}C$) and nitrogen ($\delta^{15}N$) of fresh leaves, litter and topsoils were used to characterize soil organic matter dynamics of twelve tropical ecosystems in the Mount Kilimanjaro region, Tanzania. We studied a total of 60 sites distributed along five individual elevational transects (860 – 4,550 m a.s.l.), which define a strong climatic and land use gradient encompassing semi-natural and managed ecosystems. The combined effects of contrasting environmental conditions, vegetation, soil, and management practices had a strong impact on the $\delta^{13}C$ and $\delta^{15}N$ values observed in the different ecosystems. The relative abundance of $C_3$ and $C_4$ plants greatly determined the $\delta^{13}C$ of a given ecosystem. In contrast, $\delta^{15}N$ values were largely controlled by land-use intensification and climatic conditions.

The large $\delta^{13}C$ enrichment factors ($\delta^{13}C_{litter} - \delta^{13}C_{soil}$) and low soil C/N ratios observed in managed and disturbed systems agree well with the notion of altered SOM dynamics. Besides the systematic removal of plant biomass characteristic of agricultural systems, annual litterfall patterns may also explain the comparatively lower contents of C and N observed in the topsoils of these intensively managed sites. Both $\delta^{15}N$ values and calculated $\delta^{15}N$-based enrichment factors ($\delta^{15}N_{litter} - \delta^{15}N_{soil}$) suggest tightest nitrogen cycling at high-elevation (>3,000 m a.s.l.) ecosystems, and more open nitrogen cycling both in grass-dominated and intensively managed cropping systems. However, claims about the nature of the N cycle (i.e. open/close) should not be made solely on the basis of soil $\delta^{15}N$ as other processes that barely discriminate against $^{15}N$ (i.e. soil nitrate leaching) have been shown to be quite significant in Mt Kilimanjaro's forest ecosystems. The negative correlation of $\delta^{15}N$ values with soil nitrogen content and the positive correlation with mean annual temperature suggest reduced

mineralisation rates, and thus limited nitrogen availability, at least in high-elevation ecosystems. By contrast, intensively managed systems are characterized by lower soil nitrogen contents and warmer conditions, leading together with nitrogen fertilizer inputs to lower nitrogen retention, and thus, significantly higher soil $\delta^{15}N$ values. A simple function driven by soil nitrogen content and mean annual temperature explained 68 % of the variability in soil $\delta^{15}N$ values across all sites. Based on our results, we suggest that in addition to land use intensification, increasing temperatures in a changing climate may promote soil carbon and nitrogen losses, thus altering the otherwise stable soil organic matter dynamics of Mt. Kilimanjaro's forest ecosystems.

**1 Introduction**

Conversion of natural ecosystems to agriculture is a worldwide phenomenon, which is of particular significance in tropical regions where human population growth rates are currently the highest (FAO and JRC, 2012). Changes in climate and land-use significantly alter vegetation composition and biogeochemical cycles, causing a strong impact on carbon (C) and nitrogen (N) turnover and stocks (Smith et al., 2014). Tropical forest biomes are particularly relevant in this context, as they are significant C storages and N turnover hotspots (Bai et al., 2012; Hedin et al., 2009; Lewis et al., 2009; Pan et al., 2011; Vitousek, 1984). Considering the increasing pressure on natural land, it gets even more crucial to understand how anthropogenic interventions affect ecosystem C and N cycling, and gain better knowledge about the main drivers of nutrient cycling, and associated exchange processes with the atmosphere and hydrosphere in tropical environments.

Research exploiting the natural abundance of stable isotopes has proved quite suitable for investigating potential impacts of land-use and/or climate change on C and N cycling in terrestrial systems (Michener and Lajtha, 2007; Pannetieri et al., 2017; Saiz et al., 2015a). Variations in the stable isotopic composition of C ($\delta^{13}C$) and N ($\delta^{15}N$) in plants and soils are the result of fractionation processes occurring during ecosystem exchange of C and N. Thus, $\delta^{13}C$ and $\delta^{15}N$ can serve as valuable indicators about ecosystem state and provide useful insights on how these systems respond to biotic and abiotic factors (Dawson et al., 2002; Högberg, 1997; Ma et al., 2012; Pardo and Nadelhoffer, 2010; Peterson and Fry, 1987; Robinson, 2001).

Plants discriminate against $^{13}CO_2$ (carbon dioxide) during photosynthetic $CO_2$ fixation depending on plant metabolism (i.e. $C_3$ and $C_4$ photosynthetic pathways). Most tropical grasses typically employ the $C_4$ photosynthetic pathway ($\delta^{13}C$ values >-15 ‰), while trees and shrubs use the $C_3$ photosynthetic pathway ($\delta^{13}C$ values <-24 ‰) (Bird et al., 1994; Bird and Pousai, 1997; Cernusak et al., 2013; Farquhar et al., 1980). The distribution of $C_3$ and $C_4$ vegetation show clear patterns along elevational gradients, with increasing abundance of $C_3$ species towards high elevations (Bird et al., 1994; Körner et al., 1991; Tieszen et al., 1979). Environmental conditions such as water availability also exert a significant influence on isotopic discrimination during atmospheric $CO_2$ fixation. Accordingly, compared to optimal moisture conditions, water stress leads to

enrichment of $^{13}$C in C$_3$ plants (Farquhar and Sharkey, 1982), while this isotopic fractionation is less obvious or even absent
in C$_4$ plants (Ma et al., 2012; Swap et al., 2004).
The soil organic matter (SOM) pool integrates the isotopic signature of the precursor biomass over different spatiotemporal
scales (Saiz et al., 2015a). Variation in soil $\delta^{13}$C values represents a valuable tool to better assess SOM dynamics,
mineralisation processes, or reconstruct past fire regimes (Saiz et al., 2015a; Wynn and Bird, 2007). The $\delta^{13}$C of SOM in a
given ecosystem is greatly controlled by the relative abundance of C$_3$ and C$_4$ plants due to their contrasting C isotopic
composition. Therefore, strong variations in soil $\delta^{13}$C can also be used to identify sources of particulate organic matter as
well as vegetation shifts such as woody thickening. However, fractionation effects associated to differential stabilisation of
SOM compounds, microbial re-processing of SOM, soil physico-chemical characteristics, and the terrestrial Seuss effect
preclude a straightforward interpretation of soil $\delta^{13}$C values (Saiz et al., 2015a).
Plant and soil $\delta^{15}$N relate to environmental and management conditions controlling N turnover, availability, and losses. $\delta^{15}$N
values of soils are generally more positive than those of vegetation due to the relatively large isotopic fractionation occurring
during soil N transformations (Dawson et al., 2002). The N-cycle of a given ecosystem may be characterized as closed, if
both efficient microbial N retention and absence of external N-inputs (e.g. atmospheric deposition and fertilizer additions)
prevent substantial gaseous and/or leaching N-losses. In contrast, open ecosystem N-cycling is characterized by significant
inputs and losses of N. On the one hand, gaseous N losses from soils are strongly depleted in $^{15}$N due to the high
fractionation factors associated to these processes (Denk et al., 2017). This results in high $\delta^{15}$N values of the residual
substrate, which consequently leaves less importance to impacts of external N additions (Robinson, 2001; Zech et al., 2011).
On the other hand, N leaching seems to only discriminate slightly against ecosystem $^{15}$N. According to Houlton and Bai
(2009) $\delta^{15}$N values of drained water agree well with those of soils across various natural ecosystems worldwide. Moreover, it
is also important to consider that soil $\delta^{15}$N may also be influenced by other factors including rooting depth, uptake of
different N compounds, and symbiotic N$_2$-fixation (Nardoto et al., 2014). Variations in $\delta^{15}$N values of plants and soils have
been successfully applied to characterize N cycling across a large variety of ecosystems worldwide (Amundson et al., 2003;
Booth et al., 2005; Craine et al., 2015a, 2015b; Martinelli et al., 1999; Nardoto et al., 2014). This includes research work that
has particularly focused on the study of N-losses derived from land-use changes or intensification (Eshetu and Högberg,
2000; Piccolo et al., 1996; Zech et al., 2011).
Information on ecosystem C and N cycling is still scarce in many tropical ecosystems, particularly in remote regions of
Africa (Abaker et al., 2016; 2018; Saiz et al., 2012; Townsend et al., 2011). Furthermore, feedbacks between C and N cycles
such as limitations of N availability in ecosystem C sequestration and net primary productivity of tropical forest require
urgent investigations (Gruber and Galloway, 2008; Zaehle, 2013). In such context, the Kilimanjaro region in Tanzania offers

the rare possibility to study a broad range of tropical ecosystems across contrasting land-use management intensities and varying climatic conditions. This region hosts a large variety of semi-natural and managed ecosystems ass a result of the strong elevational and land-use gradient.

We hypothesized that (i) vegetation composition ($C_3$/$C_4$) is the main control for ecosystem $\delta^{13}C$ values, whereas (ii) $\delta^{15}N$ values are rather controlled by land use management and climatic conditions. The main aim of this study is to evaluate the potential of $\delta^{13}C$ and $\delta^{15}N$ values in plant and soil material to assess C and N cycling across a broad variety of semi-natural and managed ecosystems under varying climatic conditions.

## 2 Materials and Methods

### 2.1 Study Sites

This study was conducted on the southern slopes of Mount (Mt.) Kilimanjaro (3.07° S, 37.35° E, 5,895 m a.s.l.) in North-East Tanzania. The climate is characterized by a bimodal precipitation pattern with a major rainy season between March and May, and the other peak between October and November. Recently, Appelhans et al. (2016) used a network of 52 meteorological stations strategically deployed in the Kilimanjaro region to measure air temperature and precipitation. They then used geo-statistical and machine-learning techniques for the gap filling of the recorded meteorological time series and their regionalization, which provides the means to calculate the meteorological data used for the complete set of sites (60) used in our work. Please refer to Appelhans et al. (2016) for more details. Maximum mean annual precipitation (MAP) of 2,552 mm occurs at an elevation of around 2,260 m a.s.l., decreasing towards lower as well as higher elevations, reaching 657 and 1,208 mm y$^{-1}$ at 871 and 4,550 m respectively (Table 1). Variations in air temperature are dominated by diurnal rather than seasonal patterns (Duane et al., 2008). Mean annual temperature (MAT) decreases with increasing elevation, ranging from 24.8 °C at 860 m to 3.5 °C at 4,550 m (Table 1).

Five altitudinal transects ranging from 860 to 4,550 m a.s.l. were established along the mountain slopes. At each transect, twelve ecosystems occurring over a strong land use gradient encompassing intensively managed cropping systems and semi-natural stands were investigated. Hence, the total number of plots studied was 60 (5 transects x 12 ecosystems; Table 1 and Fig. 1). The cropping systems comprised multi-layer and multi-crop agroforestry homegardens (Hom), monoculture coffee plantations (Cof) with dispersed shading trees, and maize fields (Mai) subject to regular albeit moderate fertilizer and pesticide applications. Plant litter is regularly removed from Cof and Mai sites. Homegardens are manually ploughed, while combustion engine machinery is used for ploughing coffee plantations and maize fields. Coffee plantations are irrigated with drip irrigation systems. Both Hom and Cof sites still host indigenous forest trees that include *Albizia schimperi*, a species that may potentially fix atmospheric N. This is one of the 5 most abundant species in 2 and 4 of the Hom and Cof sites respectively, making up less than 25% of the vegetation cover in all cases. Grasslands (Gra) and savannas (Sav) are

extensively managed by means of domestic grazing and occasional grass cutting, thus having significantly lower anthropogenic disturbances than cropping systems. Semi-natural ecosystems include several montane forest stands. These include lower montane (Flm), *Ocotea* (Foc), *Podocarpus* (Fpo), *Erica* (Fer), and alpine shrub vegetation *Helichrysum* (Hel). Even though lower montane forests are currently under protection they are still subject to sporadic illegal logging. In addition to sampling undisturbed forest ecosystems of *Ocotea* and *Podocarpus*, we purposely studied sites that had been affected by logging activities and fire events prior to the establishment of the Kilimanjaro National Park (Soini, 2005): *Ocotea* (Fod) and *Podocarpus* (Fpd) (Table 1). Erica forests represent Africa's highest forests in the subalpine zone. Higher above is the alpine zone, the realm of Helichrysum vegetation that is dominated by cushion plants and tussock grasses (Ensslin et al., 2015; Hemp, 2006). Potential ecosystem productivity and decomposition rates show a hump-shaped pattern resembling that of precipitation (Fig S1). It is interesting to see the close match between the two variables along the elevation range, albeit this trend weakens slightly towards higher elevation sites. Optimum growth and decomposition conditions are shown between 1,800 and 2,500 m.a.s.l.. These locations correspond to low altitude forest ecosystems (Flm and Foc) that do not experience severe seasonal limitations in moisture or temperature as it is otherwise the case in lower as well as higher elevation systems that are moisture and temperature limited respectively (Becker and Kuzyakov, 2018).

Detailed physico-chemical characteristics of the dominant soils are listed in Table 1. Soils in the Mt. Kilimanjaro region are mainly derived from volcanic rocks and ashes. The wide array of climatic conditions present along the elevational gradient influence soil genesis, which results in the occurrence of andosols at high elevations, and soils of more advanced genesis at lower elevations (e.g. nitosols) (Majule, 2003).

It is extremely difficult to provide reliable estimates of both fertilizers and pesticide rates used in small household farms in sub-Saharan Africa. This is because the actual use of these products is strongly dependent on both its availability in the local/regional market, the economic circumstances of each individual farmer, and individual perceptions about their use (Saiz and Albrecht, 2016). The only sites receiving fertilizer are the two monocultures: Maize (Mai) fields and Coffee (Cof) plantations, and to a lesser extent the homegardens (Hom) sites. In the latter sites Gütlein et al. (2018) report that weed control is mainly done by hand, and the use of mineral or organic N-fertilizers is low or non-existent. Extensively managed sites (i.e. Sav and Gra) receive varying amounts of organic inputs as a result of grazing activities, but again, their actual rates are unknown. A more detailed explanation on fertilizer and pesticides inputs used in the region is provided in the Supplementary Information.

**2.2 Sampling and Analyses**

Fieldwork took place in February and March in 2011 and 2012. Sampling was conducted on 50 x 50 m plots established at each of the 60 studied sites (12 ecosystems x 5 transects). Surface litter and mineral topsoil (0-5 cm) were sampled at five

locations (four corners and the central point) at each plot. Additionally, fresh mature leaves of the five most abundant plant
species covering 80% of total plant biomass per site were collected (Schellenberg Costa et al., 2017). All sampled materials
(leaves, litter and soil) were air-dried until constant weight, and leaf material was subsequently oven-dried at 70 °C for 60
hours prior to grinding. Soil was sieved to 2 mm with visible root fragments being further removed prior to grinding with a
mixer mill (MM200, Retsch, Haan Germany). Soil pH was determined with a pH meter (Multi Cal SenTix61, WTW,
Weilheim, Germany) in a 0.01 M $CaCl_2$ solution, with a $CaCl_2$ to soil ratio of 2:1. Particle size distribution was determined
gravimetrically using the pipette method (van Reeuwijk, 2002).
All soil, litter, and leaf samples were analysed with a dry combustion elemental analyzer (Costech International S.p.A.,
Milano, Italy) fitted with a zero-blank autosampler coupled to a ThermoFinnigan DeltaPlus-XL using Continuous-Flow
Isotope Ratio Mass Spectrometry (CF-IRMS) for determination of abundance of elemental C and N, and their stable isotopic
composition ($\delta^{13}C$, $\delta^{15}N$). Precisions (standard deviations) on internal standards for elemental C and N concentrations and
stable isotopic compositions were better than 0.08 % and 0.2 ‰ respectively.
Natural $^{13}C$ or $^{15}N$ abundances are expressed in $\delta$ units according to Eq. (1):
$\delta$ (‰) = ($R_{sample}$ - $R_{standard}$ / $R_{standard}$) x 1000,                                              (1)
where $R_{sample}$ denotes the ratio $^{13}C/^{12}C$ or $^{15}N/^{14}N$ in the sample, and $R_{standard}$ denotes the ratios in Pee Dee Belemnite or
atmospheric $N_2$ (international standards for C and N, respectively). The average values for the plant samples were weighted
considering their relative abundance at each site. Individual values for soil, litter, and leaves were averaged for each plot.
In addition, both $\delta^{13}C$- and $\delta^{15}N$-based enrichment factors ($\varepsilon$) were calculated following Eqs. 2 and 3:
$\varepsilon_C = \delta^{13}C_{litter} - \delta^{13}C_{soil}$,                                              (2)
$\varepsilon_N = \delta^{15}N_{litter} - \delta^{15}N_{soil}$,                                              (3)
These were used as indicators for SOM decomposition dynamics and ecosystem N status (Garten et al., 2008; Mariotti et al.,
1981). Note that we use the stable isotopic values values of litter material rather than fresh leaves from various species to
calculate enrichment factors, since litter provides a more unbiased representation of the quality, quantity, and spatiotemporal
dynamics of organic inputs entering the SOM pool (Saiz et al., 2015a).
**2.3 Statistical Analysis**
Normal distribution of the data was confirmed with the Shapiro-Wilk test. One-way ANOVA was performed to test for
significant differences between ecosystems, while Tukey's HSD was used as post hoc procedure to test for significant
differences across sites ($P \leq 0.05$). Correlation analyses were performed to identify soil, foliar, and climatic variables

176 influencing soil $\delta^{15}N$ values. Subsequently, a principal component analysis (PCA) was conducted to reveal relationships

177 between the main variables affecting soil $\delta^{15}N$ values. The PCA was based on a correlation matrix including soil (C and N

178 concentrations, C/N ratio, $\delta^{13}C$, pH values, sand and clay contents) as well as climatic parameters (MAT and MAP). A

179 stepwise multiple regression was used to identify the main driving parameters determining soil $\delta^{15}N$ across the elevational

180 transect. All statistical analyses were conducted with R (version 3.2.2; R Core Team, 2015).

181 **3 Results**

182 **3.1 General soil characteristics**

183 Soil C and N contents were the highest in forest ecosystems and showed a decreasing trend towards managed sites (i.e.

184 homegardens, grasslands, coffee and maize fields) (Table 1). Also, natural savannas and *Helichrysum* ecosystems had lower

185 soil C and N values compared to forest ecosystems. The low temperatures and sandy nature of the *Helichrysum* sites play a

186 strong role in their characteristically low productivity and moderate decomposition potentials (Table 1; Fig. S1), which

187 unquestionably affects the comparatively low soil C and N contents of these alpine systems.

188 An opposite trend to that of soil C and N abundance was observed for soil C/N ratios, whereby managed sites showed

189 significantly lower values compared to those of semi-natural ecosystems. Soil pH values revealed acidic conditions at all

190 sites, with the lowest values observed in forest sites having comparatively higher MAP (Table 1).

191 **3.2 Variation of $\delta^{13}C$ values along the elevational and land-use gradient**

192 There were large variations in $\delta^{13}C$ values along the elevational and land-use gradient, with distinct differences between

193 managed and semi-natural ecosystems (Fig. 2). Compared to soils and litter, leaves invariably showed the lowest $\delta^{13}C$ values

194 in all the studied ecosystems, with the exception of grasslands and savannas that exhibited lower soil $\delta^{13}C$ values than plant

195 material.

196 The $\delta^{13}C$ values of semi-natural ecosystems ranged between -32.8 and -24.1 ‰ (mean ± SE: soil -26.0 ± 0.2 ‰; litter -27.2 ±

197 0.2 ‰; leaves -29.3 ± 0.3 ‰), showing a progressive reduction with decreasing elevation (i.e. from 4.500 to 1,750 m a.s.l.;

198 Fig. S2). The variation in $\delta^{13}C$ values was much higher (-29.7 to -13.3 ‰) in managed ecosystems located at lower

199 elevations (i.e. between 860 and 1,750 m a.s.l.; Fig. S2). The highest $\delta^{13}C$ values were observed in $C_4$-dominated ecosystems

200 (i.e. savannas, maize fields, and grasslands; soil -16.8 ± 0.6 ‰, litter -19.3 ± 0.8 ‰, leaves -18.8 ± 1.1 ‰); while lower $\delta^{13}C$

201 values were obtained for coffee plantations and homegardens (soil -24.8 ± 0.5 ‰, litter -27.2 ± 0.4 ‰, leaves -27.3 ± 0.4 ‰).

202 Coffee plantations showed a slight influence of $C_4$ vegetation in the soil data as a result of grasses growing between the rows

203 of coffee plants. No significant variations were observed between $\delta^{13}C$ values of soils and those of litter and leaves in the

ecosystems with predominance of $C_4$ vegetation (savannas, maize fields and grasslands). Exploratory data analyses revealed that in most cases, soil, litter, leaf, and climatic variables cross-correlated with each other (Table S1).

Figure 3 shows relatively small variations in $\delta^{13}C$ enrichment factors (> -1.25 ‰) both in undisturbed semi-natural and extensively managed sites along the elevational gradient, while managed and disturbed sites show higher and more variable $\delta^{13}C$ enrichment factors.

**3.3 Variation of $\delta^{15}N$ values along the elevational and land-use gradient**

Significantly higher $\delta^{15}N$ values were observed for all sampled materials in the intensively managed (cropping) systems compared to semi-natural and grass-dominated ecosystems (Fig. 4a). The $\delta^{15}N$ values for managed systems ranged between -2.6 and 7.8 ‰ (mean ± SE: soil 5.6 ± 0.3 ‰, litter 1.7 ± 0.5 ‰, leaves 2.0 ± 0.5 ‰). By contrast, semi-natural ecosystems had considerably lower $\delta^{15}N$ values, which ranged from -5.0 to 3.6 ‰ (soil 1.5 ± 0.2 ‰, litter -2.1 ± 0.2 ‰, leaves -1.3 ± 0.3 ‰). Soil $\delta^{15}N$ values were significantly higher than those of leaves and litter across all the ecosystems studied, with the only exception of agroforestry homegardens (Fig. 4a). $\delta^{15}N$ values of leaves and litter did not show significant differences within any given ecosystem.

Calculated $\delta^{15}N$-based enrichment factors showed high variability across all ecosystems with values ranging from -7.5 to -1.6 ‰ (Fig. 4b). A differentiation between managed and natural ecosystems was less clear than for $\delta^{15}N$ values. The most negative enrichment factors (< -4.0 ‰) were observed for *Helichrysum, Erica, Podocarpus* disturbed, and grass-dominated ecosystems (savannas and grasslands). These enrichment factors were significantly less negative for montane forests at lower elevations (*Podocarpus*, *Ocotea* and lower montane) and intensively managed (cropping) systems (i.e. homegarden, coffee, and maize; Fig. 4b).

**3.4 Impacts of soil and climatic variables on soil $\delta^{15}N$ values**

Two principal components (PC) explained 78.3 % of the total soil $\delta^{15}N$ variation (Fig. 5). The first component explained 55.8 % of the variability, and included soil chemistry and climatic variables (soil C and N concentrations, soil C/N ratio, soil pH, soil $\delta^{13}C$, MAP and MAT). Highly significant correlations ($P < 0.001$) were obtained between PC 1 and the above factors (r = 0.93, 0.93, 0.61, -0.87, -0.76, 0.87, and -0.63, respectively; Table S2). The second component explained an additional 22.5 % of soil $\delta^{15}N$ variability and included soil texture (clay and sand contents) and MAT. These variables were highly correlated with PC 2 (r = -0.84, 0.82, and -0.65; Table S2). The principal component bi-plot showed a strong grouping between managed and semi-natural ecosystems (Fig. 5). Managed sites clustered around MAT, soil $\delta^{13}C$, and soil pH, while $C_4$-dominated ecosystems (grassland, savannas, and maize fields) were preferentially influenced by the latter two variables.

In contrast, semi-natural montane forest ecosystems, rather grouped around soil chemical properties such as C and N contents, C/N ratio, as well as MAP, while alpine *Helichrysum* ecosystems clustered around soil sand content.

In addition to PCA, multiple regression analyses were performed using a stepwise procedure that identified soil N content and MAT as the main driving variables explaining the variation in soil $\delta^{15}N$. A paraboloid model explained 68 % of this variability ($P < 0.05$; Fig. 6). The combination of relatively high soil N contents (1 to 3 %), and low MAT (up to 14 °C), invariably corresponded to low soil $\delta^{15}N$ values (< 2 ‰) characteristic of semi-natural ecosystems. Conversely, the relatively high soil $\delta^{15}N$ values (> 2 ‰) observed in managed ecosystems corresponded to low soil N contents (<1 %) and comparatively high MAT (17 to 25 °C).

The relationship between soil $\delta^{15}N$ values and climatic and edaphic variables provided valuable information about potentially different SOM dynamics in the various ecosystems studied, with data showing a clear differentiation between semi-natural and managed ecosystems (Fig. S4). The former is characterized by comparatively higher C/N ratios and lower $\delta^{15}N$ values (averaging 15.5 and 1.5 ‰ respectively), while the latter showed lower C/N ratios and higher soil $\delta^{15}N$ values (averaging 11.9 and 3.5 ‰ respectively). Managed ecosystems further grouped into intensively cropped (homegardens, maize fields, and coffee plantations) and extensively managed grass-dominated ecosystems (savannas and grasslands).

**4 Discussion**

**4.1 Factors influencing the variation of $\delta^{13}C$ values along the elevational and land-use gradient**

The $\delta^{13}C$ values of leaves in $C_3$-dominated (semi-natural) ecosystems in Mt. Kilimanjaro increased with elevation (Figs. 1 and S2), which is in agreement with findings from other mountainous ecosystems in the tropics, Europe, and North America (Bird et al., 1994; Körner et al., 1991; Ortiz et al., 2016; Zhou et al., 2011; Zhu et al., 2009). The wider scatter of $\delta^{13}C$ values observed in leaves relative to soils is most certainly due to the inherently large (inter- and intra- specific) variability of $\delta^{13}C$ in plants (Bird et al., 1994). Different tissues within the plant can present widely divergent $\delta^{13}C$ values as a result of fractionation processes associated with the C compounds involved in their construction (Dawson et al., 2002). Moreover, other factors including light intensity, humidity, and the re-utilization of previously respired low $^{13}C$-$CO_2$ within the canopy may further contribute to the variability of $\delta^{13}C$ in leaf tissues (Ometto et al., 2006; van der Merwe and Medina, 1989).

While fractionation effects preclude a straightforward interpretation of $\delta^{13}C$ of SOM, this variable provides an integrated measure of the isotopic composition of the precursor biomass at the ecosystem level (Bird et al., 2004; Saiz et al., 2015a). Mass balance calculations that assume (i) 5% (w/w) average root mass (< 2 mm) in soil samples, and (ii) leaves having similar isotopic signals as roots, show that the removal of visible sieved roots might cause a very small effect on soil isotopic values. This would amount to values ~0.15‰ higher than the original soil isotopic values, with such discrepancy being even

smaller if root samples were considered having values 0.5-1‰ higher than leaves as is commonly reported in the literature (calculations not shown). Besides the natural variability of soil $\delta^{13}C$ values observed in $C_3$-dominated semi-natural ecosystems, there were distinct patterns in $\delta^{13}C$ values of soil samples collected in extensively managed, low-elevation ecosystems where woody and grass vegetation coexist (i.e. grasslands and savannas), which indicate the strong influence exerted by $C_4$ vegetation on the C isotopic composition of all sampled materials (Fig. 2). The results obtained in semi-natural ecosystems at Mt. Kilimanjaro fit well within the interpretative framework for elevational soil $\delta^{13}C$ data proposed by Bird et al. (1994). These authors suggest that besides temperature and atmospheric pressure, other primary factors influencing soil $\delta^{13}C$ values are the age and degree of decomposition of SOM, as well as variables related to the characteristics of the canopy, including the proportion of respired $CO_2$ that is recycled during photosynthesis, the relative contribution of leaf and woody litter to SOM, and soil moisture.

Besides the factors explained above, soil $\delta^{13}C$ values are strongly influenced by the balance between ecosystem C inputs and outputs. It seems reasonable to assume that in the case of natural ecosystems there may be a steady state between SOM inputs and decomposition rates. This should be in contrast with the typically altered nutrient dynamics of disturbed systems, particularly those under agricultural management (Wang et al., 2018). Low fractionation factors in $\delta^{13}C$ are commonly reported between plant material and topsoils in natural systems mainly because of the relatively limited humification of recent organic matter prevalent in topsoils (Acton et al., 2013; Wang et al., 2018). Thus, we hypothesized that if C inputs and outputs were roughly in balance, then the difference in $\delta^{13}C$ values between plant material and topsoil would be smaller in undisturbed sites compared to managed or disturbed sites. The results shown in Fig. 3 agree well with this notion.

Soil $\delta^{13}C$ values decreased with increasing MAP and decreasing MAT, which also corresponded with higher SOC contents (Fig. S3). This suggests that the relatively cooler and wetter conditions of high elevation semi-natural forest ecosystems (i.e. Foc, Fpo) promote the accumulation of SOM, which is similar to previous findings of work conducted along elevational gradients (Bird et al., 1994; Kohn, 2010). Compared to high-elevation locations, the climatic conditions of mid-elevation ecosystems are more favourable for the activities of SOM decomposers, as these sites are consistently warmer and drier than the characteristically cool and occasionally waterlogged high-altitude ecosystems (Fig. S1; Becker and Kuzyakov, 2018; Borken and Matzner, 2009; Garten et al., 2009; Kirschbaum, 1995; Leirós et al., 1999). The comparatively high soil $\delta^{13}C$ values observed in the disturbed *Podocarpus* (Fpd) and *Erica* forest (Fer) plots may have been partly caused by recurrent fire events (Hemp, 2005) leading to reduced SOC contents and higher C/N ratios (Saiz et al., 2015a). Further variations in soil $\delta^{13}C$ values could also be related to the biochemical composition of the precursor biomass. For instance, herbaceous vegetation is pervasive at high elevations, and contains relatively low amounts of lignin – an organic compound characteristically depleted in $^{13}C$ (Benner et al., 1987). This may contribute to explain the higher $\delta^{13}C$ values observed in

plant and soil materials in alpine ecosystems dominated by *Helichrysum* vegetation, compared to forest ecosystems at lower
elevations (Fig. 2).
Elevation also has a strong influence on the seasonal litterfall dynamics observed in Mt Kilimanjaro, and thus may have
significant implications in the SOM cycling across the various ecosystems (Becker et al., 2015). These authors suggest that
the large accumulation of particulate organic matter observed at the end of the dry season in low and mid altitude ecosystems
may result in the increased mineralization of easily available substrates (Mganga and Kuzyakov, 2014) and nutrient leaching
(Gütlein et al., 2018) during the following wet season. Agricultural practices such as the removal of biomass or ploughing
deplete SOM, particularly in the intensively managed systems (i.e. maize, homegardens and coffee plantations), thus leading
to lower SOC contents and C/N ratios, and slightly higher soil $\delta^{13}$C values than those observed in semi-natural ecosystems at
comparable elevations (e.g. lower montane forests; Fig. S3). Indeed, the relationship between $\delta^{13}$C enrichment factors and
soil C/N ratios shown in Fig. 3 is quite informative regarding SOM dynamics. As previously mentioned, soil C/N ratios
provide a good indication of SOM decomposition processes, typically showing comparatively low values in managed and
disturbed systems. These correspond well with sites having large enrichment factors (< -1.25 ‰; i.e. intensively managed
and disturbed sites), which agree with the notion of altered SOM dynamics. Therefore, besides the systematic removal of
plant biomass characteristic of agricultural systems, annual litterfall patterns may also explain the comparatively lower
contents of C and N observed in the topsoils of intensively managed sites (Table 1; Figs. S3, S4). Moreover, low-elevation
ecosystems contain a variable mixture of $C_3$ and $C_4$ vegetation, which have been shown to have differential mineralization
dynamics as demonstrated by incubation experiments (Wynn and Bird, 2007), and field-based research (Saiz et al., 2015a).
Our data show strong relationships between temperature and variables directly related to SOM dynamics such as soil $\delta^{13}$C,
C, N and C/N ratios (Table S1). These results agree well with recent findings by Becker and Kuzyakov (2018) who studied
SOM decomposition dynamics at these very sites. An important finding revealed by that study is that of seasonal variation in
temperature being a major factor controlling litter decomposition. Their study shows that small seasonal variations in
temperature observed at high elevation sites exert a strong effect on litter decomposition rates. Therefore, the authors argue
that the projected increase in surface temperature may result in potentially large soil C losses at these sites due to the
comparatively strong temperature sensitivity to decomposition that is commonly observed at low temperatures and at high
elevations sites (Blagodatskaya et al., 2016).
Savannas and grasslands are subject to recurrent fire events, and thus the soils of these ecosystems may potentially contain
significant amounts of fire-derived (pyrogenic) C (Saiz et al., 2015b). This can be partly demonstrated by the higher soil C/N
ratios observed in these ecosystems compared to $C_4$-dominated agricultural systems protected from fire (e.g. maize
plantations; Fig. S3d). Moreover, the $\delta^{13}$C values of soils in grasslands and savannas were lower than those of leaves, which
may be due to the savanna isotope disequilibrium effect (SIDE) (Bird and Pousai, 1997; Saiz et al., 2015b). The latter
concept explains the difference in C isotopic composition between the precursor vegetation and pyrogenic C compounds
produced during the combustion of biomass. Saiz et al. (2015b) have demonstrated that savanna fires produce pyrogenic C
that is relatively [13]C depleted with respect to the precursor biomass. Furthermore, the combustion of $C_4$ vegetation produces
finer pyrogenic C particles than woody biomass, resulting in the preferential export of grass-derived pyrogenic particles from
the site of burning, which further enhances the depletion of [13]C in these soils (Saiz et al., 2018).

## 327 4.2 Variation of $\delta^{15}N$ values along the elevational and land-use gradient

The $\delta^{15}N$ values of leaves, litter, and topsoil presented here (Fig. 4a) agree well with the range of data reported from earlier
investigations in the same study region (Amundson et al., 2003; Zech et al., 2011), but with our study involving more
ecosystems, replicate sites and a far larger spatial sampling domain. Overall, the $\delta^{15}N$ values for montane tropical forest
ecosystems in Mt. Kilimanjaro are considerably lower than the mean values reported for a broad variety of tropical lowland
forests worldwide (soil values ranging from 3 to 14 ‰; de Freitas et al., 2015; Martinelli et al., 1999; Nardoto et al., 2014;
Piccolo et al., 1996; Sotta et al., 2008). Rather, the $\delta^{15}N$ values observed in the montane forests investigated are in the same
range of temperate forest ecosystems reported in a comprehensive literature review by Martinelli et al. (1999). These authors
argue that, compared to tropical lowland forests, the lower $\delta^{15}N$ values of temperate and montane tropical forests result from
their lower N availability and thus lower ecosystem N losses. However, this hypothesis may not completely hold for the
montane forest ecosystems of our study, since Gütlein et al. (2018) reported elevated soil $NO_3^-$ and DON concentrations at
deep soil solution (80 cm) and significant nitrogen leaching rates of 10 - 15 kg N ha$^{-1}$ y$^{-1}$. The relatively low $\delta^{15}N$-based
enrichment factors observed in the lower montane, *Ocotea* and undisturbed *Podocarpus* forest (Fig. 4b) were probably due
to the prevalence of biological di-nitrogen fixation (BNF) at these ecosystems. The assumption of significant BNF is
supported by leaf $\delta^{15}N$ values close to 0 ‰ (Fig. 4a) and is in line with previous works (Craine et al., 2015a; Nardoto et al.,
2014; Robinson, 2001). Furthermore, sporadic measurements of N-compounds in rainfall and throughfall conducted at our
forest sites showed substantial input of N via atmospheric deposition, which may be in the order of N leaching losses
(unpublished results). This agrees well with findings from Bauters et al. (2018) reporting 18 kg N ha$^{-1}$ y$^{-1}$ N inputs via wet
deposition into tropical forests of the Congo Basin, which are predominantly derived from biomass burning and long-range
atmospheric transport. High N inputs into these forest ecosystems are likely to be in a similar range as N outputs (prevailed
by leaching losses particularly where MAP is highest; Gütlein et al., 2018), and therefore, they would not translate to strong
effects on ecosystem $\delta^{15}N$ values. The significantly more negative enrichment factors observed in the disturbed *Podocarpus*
and *Erica* forests (Fig. 4b) may be related to past fire events (Hemp, 2005; Zech et al., 2011). Burning of vegetation may
cause losses of $^{15}$N-depleted NO$_\chi$ gas and N leachate, resulting in higher soil $\delta^{15}$N values, thus producing variations in $\delta^{15}$N-
based enrichment factors (Zech et al., 2011).
Previous studies have shown that $\delta^{15}$N values generally increase with land-use intensification (Martinelli et al., 1999;
Stevenson et al., 2010), which corresponds well with the more positive $\delta^{15}$N values observed in the intensively managed
agricultural systems occurring at the mountain's foot slope (Fig. 4a). Indeed, agronomic practices such as fertilization,
removal of plant material after harvest, or ploughing, are factors known to affect N turnover processes that strongly affect
$\delta^{15}$N values (Bedard-Haughn et al., 2003; Saiz et al., 2016). However, our values are in the lower range of published data for
other land-use gradients (Aranibar et al., 2008; Eshetu and Högberg, 2000; Traoré et al., 2015), and may partly be the result
of comparably low to moderate organic and inorganic N fertilization rates currently applied in the region (anecdotal evidence
gathered by the authors and SI). Additionally, the nitrogen isotopic signal of mineral fertilizers commonly used in the region
is ~0 ‰ (Bateman and Kelly, 2007), and thus, it may not exert a significant additional bias on the interpretation of soil $\delta^{15}$N
values. However, the addition of manure ($\delta^{15}$N ~8 ‰) in Hom systems, albeit used in low quantities (Gütlein et al., 2018),
may have well contributed to the high $\delta^{15}$N values observed in this ecosystem (Fig. 4). Also, we suggest that the use of
pesticides may not pose a strong bias in our isotopic results since their use is limited to intensively managed sites, and the
actual isotopic values of pesticides work in the opposite direction to the observed data (Fig. 4; SI).
Compared to other low-elevation managed stands such as homegardens and coffee plantations, the higher $\delta^{15}$N-based
enrichment factors observed in maize fields and in grass-dominated ecosystems (grasslands and savannas) (Fig. 4b) may be
related to both the organic inputs resultant from grazing activities and the influence of C$_4$ vegetation. Both Aranibar et al.
(2008) and Wang et al. (2010) have suggested that variations in $\delta^{15}$N values within a given ecosystem could be due to C$_3$ and
C$_4$ plants preferentially absorbing chemical forms of N with differing $^{15}$N abundances. Moreover, recurrent fires
characteristic of tropical grasslands and savannas may have also influenced their comparatively high soil $\delta^{15}$N, causing the
relatively high $\delta^{15}$N-based enrichment factors.
**4.3 Factors controlling soil $\delta^{15}$N along the elevational and land-use gradient**
The strong controlling effects exerted by climatic and edaphic factors on soil $\delta^{15}$N values agree well with numerous previous
works (Amundson et al., 2003; Conen et al., 2013; Eshetu and Högberg, 2000; Martinelli et al., 1999; Stevenson et al.,
2010). The principal component analysis of factors controlling soil $\delta^{15}$N revealed a strong clustering between managed and
semi-natural ecosystems (Fig. 5), which was also reflected in the multiple regression analysis and graphical representation
depicting soil $\delta^{15}$N as a function of soil N concentration and MAT (Fig. 6). Semi-natural ecosystems were characterized by
relatively low soil $\delta^{15}$N values, and occurred across a broad range of soil N contents in locations with low to medium MAT.
By contrast, intensively managed ecosystems had higher soil $\delta^{15}$N values and corresponded to locations with low soil N

contents and high MAT. The negative correlation of $\delta^{15}$N values with soil nitrogen content and the positive correlation with mean annual temperature suggest reduced mineralisation rates, and thus limited nitrogen availability, at least in high-elevation ecosystems.

The sharp contrast observed both in soil C/N ratios and $\delta^{15}$N values between managed and semi-natural ecosystems offers additional useful information about their potentially contrasting SOM dynamics (Fig. S4d). Intensively managed sites consistently showed low soil C/N ratios and high soil $\delta^{15}$N values, which may initially suggest a more open N cycle and potentially greater N losses as reported by Gerschlauer et al. (2016) for some of these ecosystems. This may due to C-limitation of heterotrophic microbial N retention under low C/N ratios (Butterbach-Bahl and Dannenmann, 2012). However, nitrate leaching is quite a relevant process that discriminates only slightly against $^{15}$N (Denk et al., 2017), which may confound the interpretation of soil $\delta^{15}$N values. Indeed, Gütlein et al. (2018) have recently shown that nitrate leaching may be quite significant in Mt Kilimanjaro's semi-natural forests. Therefore, at least in these ecosystems, claims about the nature of the N cycle (i.e. open/close) should not be made solely on the basis of soil $\delta^{15}$N.

Grass-dominated ecosystems (grasslands and savannas) were noticeably different to the intensively managed croplands, as demonstrated by the higher soil C/N ratios and lower soil $\delta^{15}$N of the former, which suggest a lower degree of decomposition of organic matter and potentially lower N turnover rates (Saiz et al., 2016). Within the intensively managed sites, the stands under maize cultivation show an interesting case of enhanced SOM dynamics. These sites are under an intensive management regime that involves the removal of aboveground vegetation after harvest. This fact combined with the faster decomposition rates reported for C$_4$-derived SOM (Saiz et al., 2015a; 2016; Wynn and Bird, 2007) may invariably lead to their characteristically low SOC and N contents (Table 1; Figs. S3, S4). Furthermore, low soil C/N ratios have been reported to enhance gaseous losses in semi-arid systems, which leads to increased soil $\delta^{15}$N values (Aranibar et al., 2004) and may explain why maize stands showed the highest soil $\delta^{15}$N values of all the land uses studied.

Semi-natural ecosystems showed rather high soil C/N ratios and low soil $\delta^{15}$N values compared to managed sites (Fig. S4d). The more humid and cooler conditions prevalent in forest ecosystems may limit decomposition processes, thereby contributing significantly to their higher SOM abundance (Table 1). A small variation range in soil $\delta^{15}$N values was also reported by Zech et al (2011) for semi-natural ecosystems (Foc and Fpo) when working along the same land-use and elevation gradient. Like us, these authors also observed a strong significant correlation of soil $\delta^{15}$N with MAT, but not with MAP (Table S1). Additionally, site-specific soil characteristics, and the structural composition of vegetation have a strong influence on ecosystem nutrient dynamics (Saiz et al., 2012; 2015a). Ecosystem disturbances (e.g. fire, selective logging, etc.) cause changes in vegetation cover that affect SOM cycling and may translate into variations in soil C/N ratios (Saiz et al., 2016). Both *Ocotea* and *Podocarpus* forests contain disturbed (Fod, Fpd) and undisturbed stands (Foc, Fpo), though only

the *Podocarpus* ecosystems allow for a general overview of disturbance impacts on SOM-related properties. While changes in the isotopic composition of C and N were not significant, soil C/N ratios were heavily influenced by disturbance (Fig. S4). Compared to non-disturbed sites, the lower C and N contents observed in the soil of disturbed ecosystems indicate reduced OM inputs to the soil and/or enhanced decomposition of SOM (Table 1). The higher soil C/N ratios observed in the *Podocarpus* disturbed and *Erica* forests may well be the result of fire, which may preferentially promote N losses while accruing relatively recalcitrant C forms (i.e. pyrogenic C). Woody biomass combustion produces pyrogenic C that accumulates preferentially close to the site of production (Saiz et al., 2018), thus likely contributing to the higher soil C/N ratios observed at these disturbed ecosystems. The lowest soil C/N ratios among all semi-natural ecosystems were observed at the alpine *Helichrysum* sites, which may relate to their characteristically sparse vegetation and extremely low MAT. Under such circumstances soil development, biomass inputs, decomposition processes, and thus, soil N turnover may be strongly limited, as it was confirmed by a recent study conducted at one of these sites (Gütlein et al., 2017).

**5 Conclusions**

The variations in $\delta^{13}$C and $\delta^{15}$N values combined with interpretation of other indices such as $\delta^{13}$C- and $\delta^{15}$N-based enrichment factors and soil C/N ratios, enabled a qualitative characterisation of regional differences in C and N dynamics as affected by vegetation characteristics, environmental conditions, and management activities.

Our data show that SOM contents are higher in cold and wet high-elevation ecosystems than at low-elevation managed sites. Management practices such as tillage, harvest, and vegetation burning promote the loss of OM, with SOM decomposition being further enhanced by the warm and moderately wet conditions of the mountain's foot slope. Based on our results, we suggest that besides management, increasing temperatures in a changing climate may promote C and N losses, thus altering the otherwise stable SOM dynamics of Mt. Kilimanjaro's forest ecosystems. Moreover, the current situation of low N inputs in managed systems of sub-Saharan Africa is likely to change, since national efforts aim to increase fertilizer use are currently <10% of recommended rates (Hickman et al., 2014). Therefore, our data may also be valuable as a generic reference for low-elevation tropical agrosystems managed under low N inputs, while it may also allow the monitoring of expected changes in agricultural management, and associated impacts on ecosystem N cycle through the study of the variation in $\delta^{15}$N values.

In addition to climatic and edaphic factors, $\delta^{15}$N values of plant and soil material can largely depend on both the amount and $\delta^{15}$N signal of atmospheric deposition and BNF, which highlights the importance of conducting additional measurements of site specific N cycling, when comparing ecosystem $\delta^{15}$N values across different biomes and regions. The combination of qualitative isotope natural abundance studies at a large number of sites (this study) with more elaborated quantitative process studies using enriched isotope labelling and N losses on a lower number of selected sites represent an ideal approach to

characterize ecosystem C and N cycling of the larger Mt. Kilimanjaro region with its diverse ecosystems, climate, and
management.

## Author contribution

FG contributed to design, performed the study, and co-wrote the paper; GS contributed to analyses and co-wrote the paper;
DSC and MK provided plant samples and contributed to writing; MD contributed to writing; and RK designed the study and
contributed to analyses and writing.

## Competing Interests

The authors declare no competing interests.

## Acknowledgments

This study was funded by the German Research Foundation (DFG: KI 1431/1-1 and KI 1431/1-2) within the Research-Unit
1246 (KiLi) and supported by the Tanzanian Commission for Science and Technology (COSTECH), the Tanzania Wildlife
Research Institute (TAWIRI) and the Mount Kilimanjaro National Park (KINAPA). In addition, the authors thank Dr.
Andreas Hemp for selection and preparation of the research plots, Prof. Dr. Bernd Huwe for correction of soil texture data,
as well as all local helpers in Tanzania, and the assistants in the laboratory of IMK-IFU in Germany. Technical support by
the Center of Stable Isotopes of KIT/IMK-IFU is gratefully acknowledged. Further thanks go to the following persons from
the KiLi project: Dr. Tim Appelhans and Prof. Dr. Thomas Nauss, Jie Zhang, Dr. Gemma Rutten, and Dr. Andreas Hemp for
providing georeferenced points underlying the GeoTIFF in Figure 1 b). We also thank three anonymous reviewers and
Jonathan Wynn for insightful comments on the MS.

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

**Table 1** General characteristics of ecosystems investigated at Mt. Kilimanjaro, Tanzania.

| Ecosystem | Land-use type | Elevation (m a.s.l.) | MAP (mm) | MAT (°C) | Soil properties | | | | | | | C/N ratio |
| | | | | | Soil type | pH (CaCl$_2$) | Clay (%) | Sand (%) | Organic carbon (%) | Total nitrogen (%) | |
|---|---|---|---|---|---|---|---|---|---|---|---|---|
| Savanna (Sav) | (M) extensive grazing, grass cutting | 971 (40) | 764 (50) | 23.7 (0.3) | Leptosol | 6.6 (0.3) | 27.3 (4.0) | 39.3 (8.7) | 3.5 (0.4) | 0.2 (0.0) | 13.5 (0.2) |
| Maize field (Mai) | (M) cropped agriculture | 938 (25) | 674 (34) | 23.6 (0.4) | Nitosol | 5.6 (0.3) | 37.4 (4.5) | 20.3 (7.7) | 1.6 (0.2) | 0.1 (0.0) | 11.8 (0.1) |
| Coffee plantation (Cof) | (M) cropped agriculture | 1,349 (78) | 1,393 (96) | 19.8 (0.7) | Vertisol | 4.5 (0.3) | 45.2 (8.0) | 17.8 (4.5) | 4.2 (0.4) | 0.4 (0.0) | 10.5 (0.2) |
| Homegarden (Hom) | (M) cropped agroforestry | 1,478 (112) | 1,656 (177) | 18.7 (0.8) | Andosol | 5.4 (0.4) | 45.4 (8.0) | 16.5 (5.8) | 6.7 (1.3) | 0.6 (0.1) | 11.5 (0.4) |
| Grassland (Gra) | (M) extensive grazing, grass cutting | 1,506 (84) | 1,610 (135) | 18.9 (0.7) | Umbrisol | 5.1 (0.4) | 48.1 (8.1) | 16.0 (5.1) | 5.3 (2.1) | 0.4 (0.2) | 12.6 (0.2) |
| Lower montane forest (Flm) | (S-N) montane forest | 1,806 (71) | 2,201 (33) | 15.5 (0.3) | Andosol | 4.7 (0.3) | 47.3 (5.2) | 14.5 (2.2) | 22.7 (4.9) | 1.6 (0.2) | 13.3 (1.5) |
| *Ocotea* forest (Foc) | (S-N) montane forest | 2,464 (106) | 2,388 (73) | 11.5 (0.4) | Andosol | 3.5 (0.2) | 52.3 (4.5) | 10.4 (2.3) | 40.2 (1.5) | 2.7 (0.1) | 14.9 (0.7) |
| *Ocotea* forest disturbed (Fod) | (S-N) montane forest | 2,378 (56) | 2,334 (35) | 11.9 (0.4) | Andosol | 3.6 (0.2) | 53.9 (3.4) | 10.1 (2.5) | 32.0 (1.8) | 2.2 (0.2) | 15.1 (1.3) |
| *Podocarpus* forest (Fpo) | (S-N) montane forest | 2,856 (41) | 2,036 (27) | 9.6 (0.2) | Andosol | 3.8 (0.1) | 48.7 (1.1) | 9.4 (1.3) | 37.0 (1.0) | 2.4 (0.1) | 15.5 (0.8) |
| *Podocarpus* forest disturbed (Fpd) | (S-N) montane forest | 2,904 (48) | 2,056 (29) | 9.7 (0.3) | Andosol | 4.0 (0.2) | 45.8 (3.4) | 12.6 (3.3) | 33.8 (2.3) | 1.7 (0.0) | 19.9 (1.4) |
| *Erica* forest (Fer) | (S-N) montane forest | 3,716 (77) | 1,517 (54) | 6.2 (0.6) | Andosol | 3.9 (0.2) | 29.5 (5.1) | 24.1 (6.2) | 28.1 (2.4) | 1.5 (0.1) | 18.9 (0.7) |
| *Helichrysum* vegetation (Hel) | (S-N) alpine scrub vegetation | 4,250 (100) | 1,293 (31) | 4.2 (0.4) | Andosol | 5.7 (0.3) | 7.9 (1.4) | 69.9 (9.5) | 6.1 (3.3) | 0.3 (0.2) | 12.0 (1.1) |

Land uses are generically classified as managed (M) and semi-natural ecosystems (S-N). MAP and MAT stand for mean annual precipitation and temperature respectively. Climatic values are according to Appelhans et al. (2016). Data represent mean values (n = 5 ± SE) for different ecosystems. The most representative soil type is shown for each ecosystem. Soil properties are given for topsoil (0 – 10 cm for pH and soil texture, 0 – 5 cm for soil organic carbon and total nitrogen).

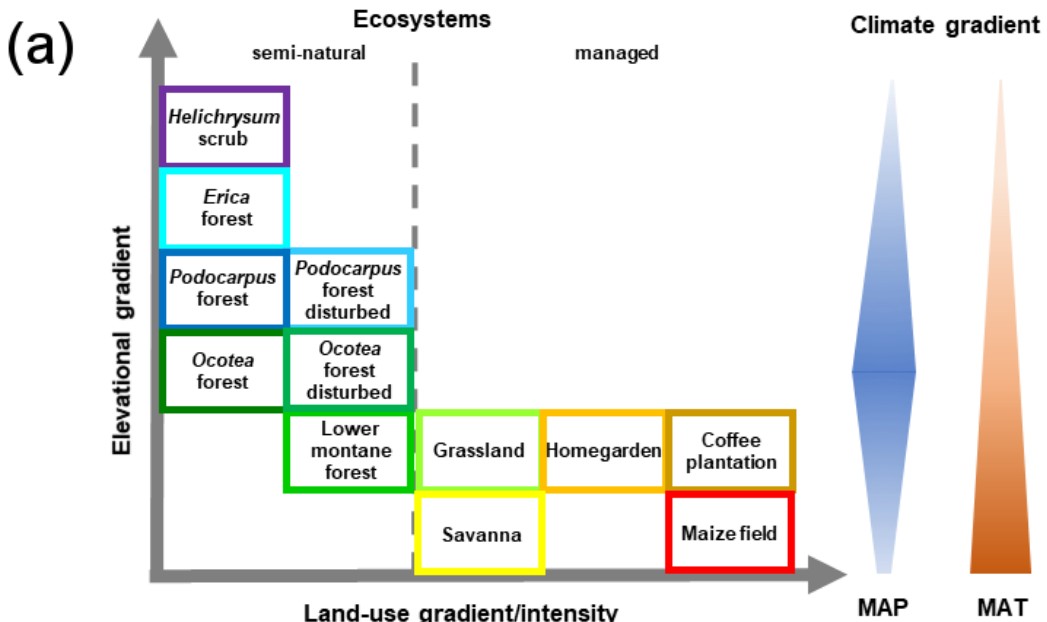

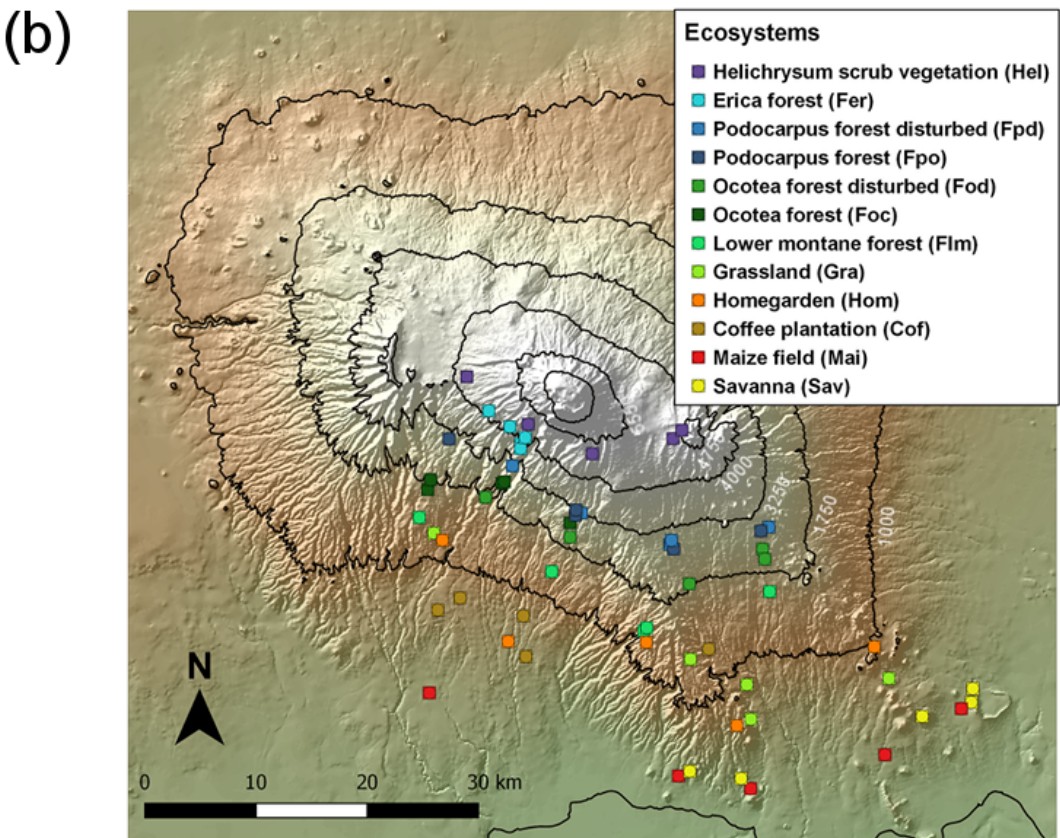

689

**Figure 1:** Geographical distribution of investigated ecosystems: a) along the elevational and land-use gradient. MAP denotes mean annual precipitation and MAT mean annual temperature. Colours of boxes framing ecosystems' names match colours of symbols in the GeoTIFF panel below; b) along the southern slope of Mt. Kilimanjaro. Symbols represent individual ecosystems (12) replicated 5 times (60 study sites in total).

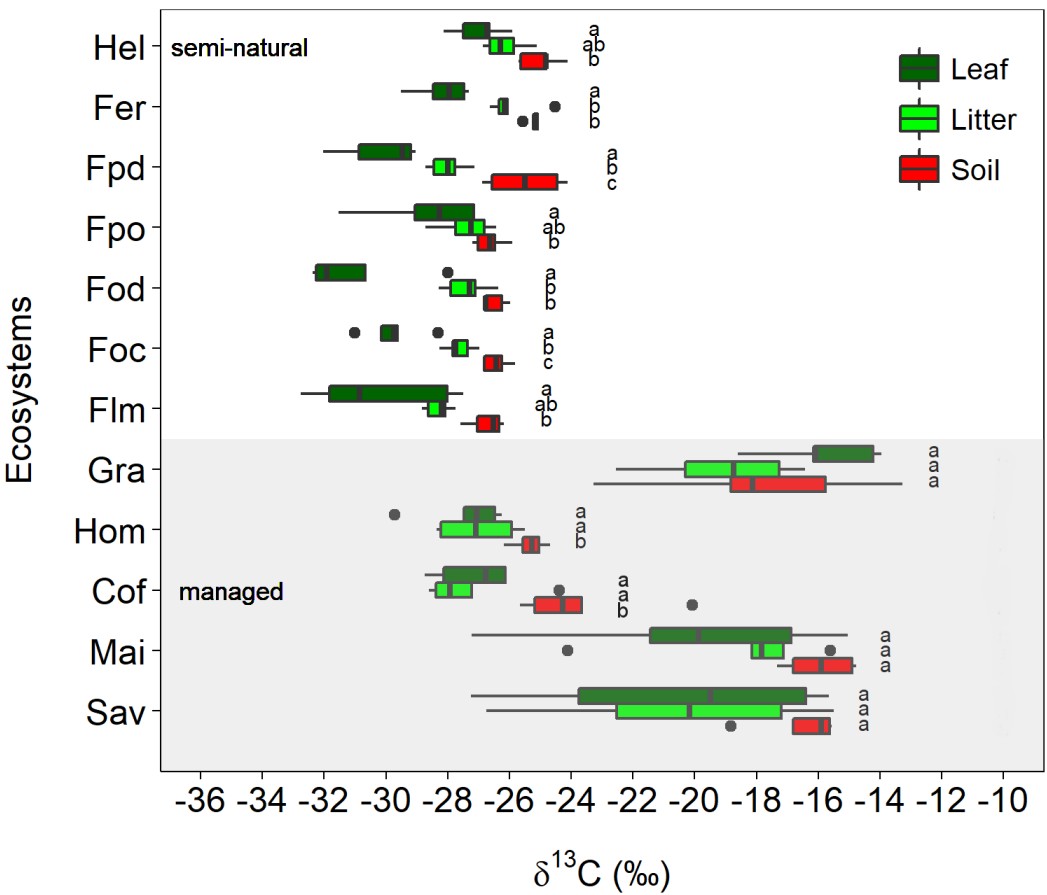

**Figure 2:** Variation in $\delta^{13}C$ values for leaves, litter, and soil along the Kilimanjaro elevational and land-use gradient. Ecosystem data represent the average values of five sites (one per each transect), with each site being composed of five samples (n = 5). Boxplots show median values per ecosystem with whiskers representing 1st and 3rd quartiles. Dots represent outliers. The shaded region represents managed ecosystems (both intensively and extensively), while those un-shaded indicate semi-natural ecosystems. Lower case letters show significant differences between sampled materials within each ecosystem (one-way ANOVA followed by Tukey's HSD test as a post hoc procedure, $P \leq 0.05$). The ecosystem acronyms used are as per Table 1. Mai, Cof, and Hom are managed cropping sites, Gra and Sav are extensively managed grasslands and savannas, while the rest represent semi-natural ecosystems. Sites are ordered by increasing altitude.

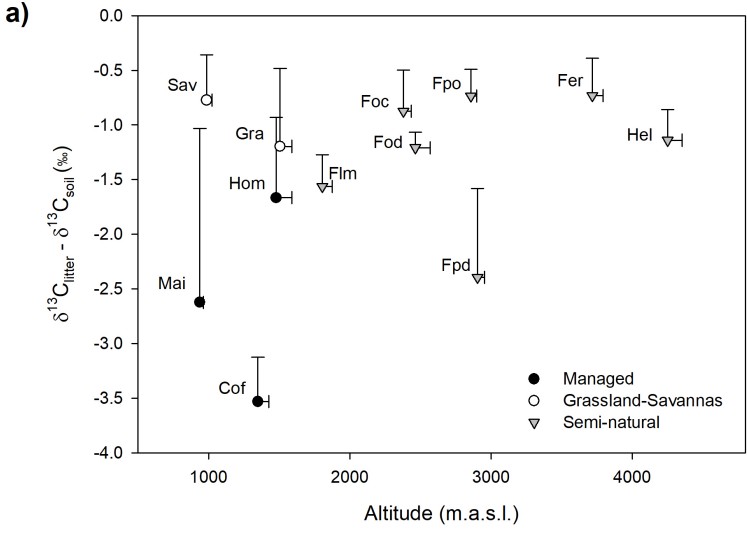

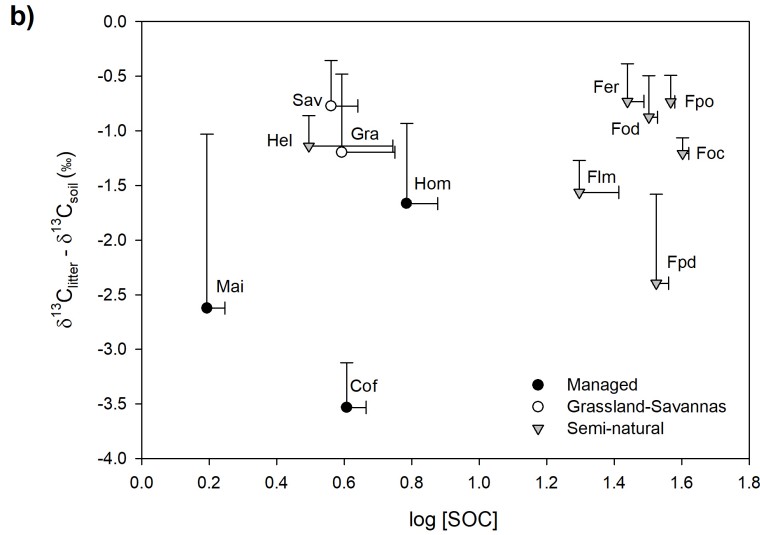

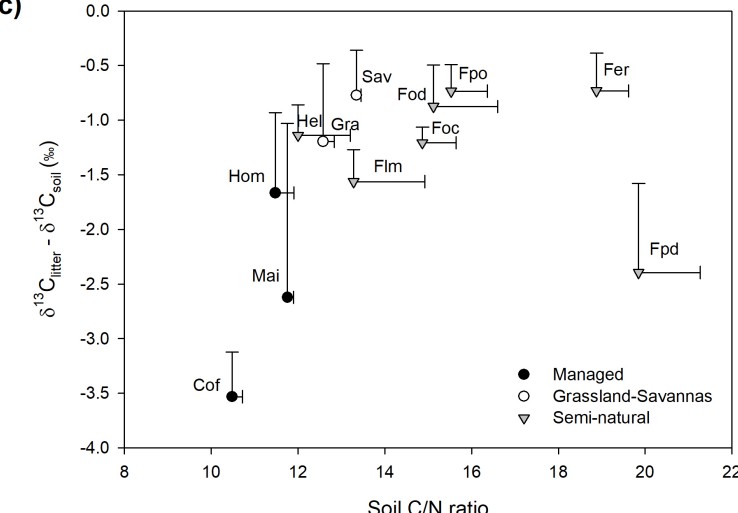

704

**Figure 3:** a) Variation in $\delta^{13}C$-based enrichment factors ($\delta^{13}C_{itter-soil}$) with elevation; b) Relationship between $\delta^{13}C$-based enrichment factors ($\delta^{13}C_{itter-soil}$) and SOC concentration (log SOC); and c) Relationship between $\delta^{13}C$-based enrichment factors ($\delta^{13}C_{itter-soil}$) and soil C/N ratios. Note: A savanna site with large $C_3$ influence was removed from the figure for clarity.

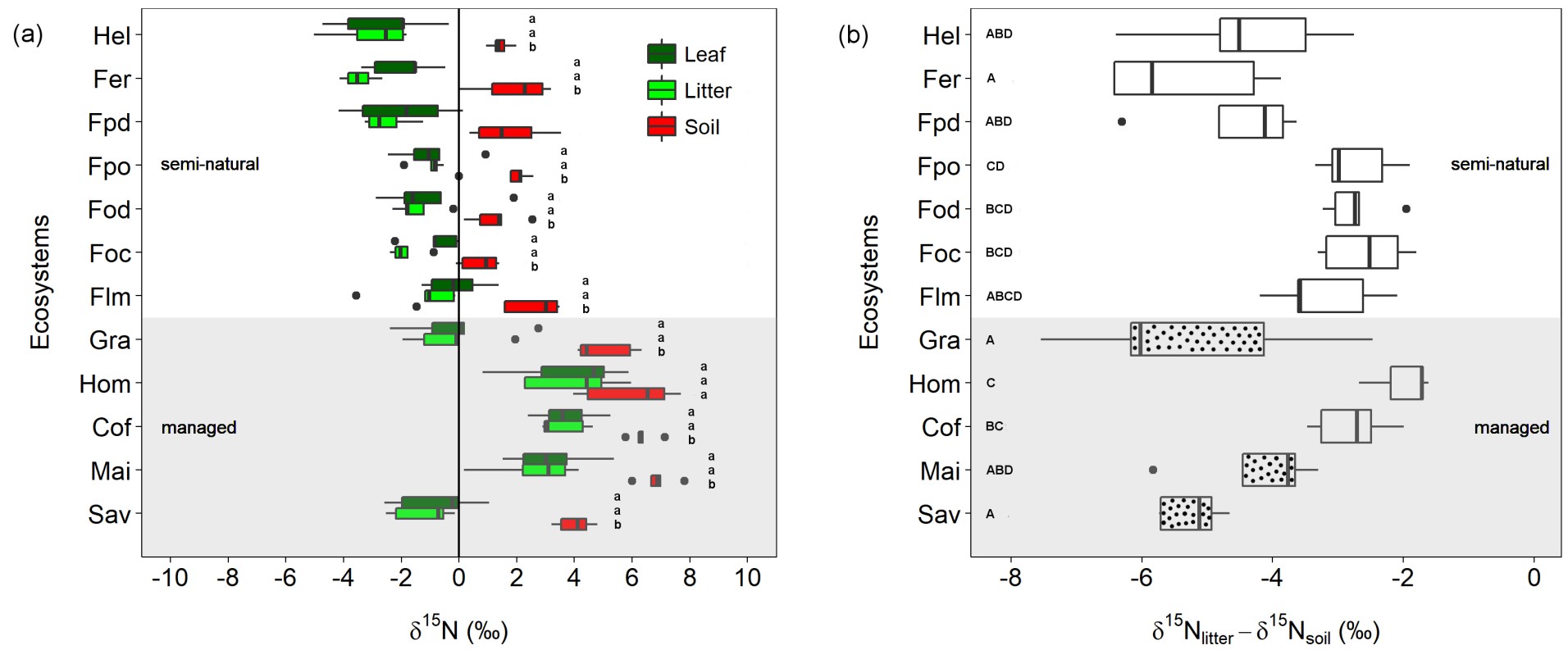

**Figure 4:** Variation in $\delta^{15}N$ values and $\delta^{15}N$-based enrichment factors along the Kilimanjaro elevational and land-use gradient. a) Variation in $\delta^{15}N$ values for leaves, litter, and soil material sampled along the Kilimanjaro elevational and land-use gradient. Boxplots show median values per ecosystem with whiskers representing 1$^{st}$ and 3$^{rd}$ quartiles. Dots represent outliers. Ecosystem data represent the average values of five sites (one per each transect), with each site being composed of five samples. Lower case letters show significant differences between sampled materials within each ecosystem (one-way ANOVA followed by Tukey's HSD test as a post hoc procedure, P ≤ 0.05); b) Variation in $\delta^{15}N$-based enrichment factors ($\delta^{15}N_{litter-soil}$) calculated for the different ecosystems along the elevational and land use gradient. Dotted boxplots indicate ecosystems dominated by C$_4$ vegetation. Capital letters indicate significant differences between ecosystems (one-way ANOVA followed by Tukey's HSD test as a post hoc procedure, P ≤ 0.05). The ecosystem acronyms used are the same as those in Table 1. Sites are ordered by increasing altitude.

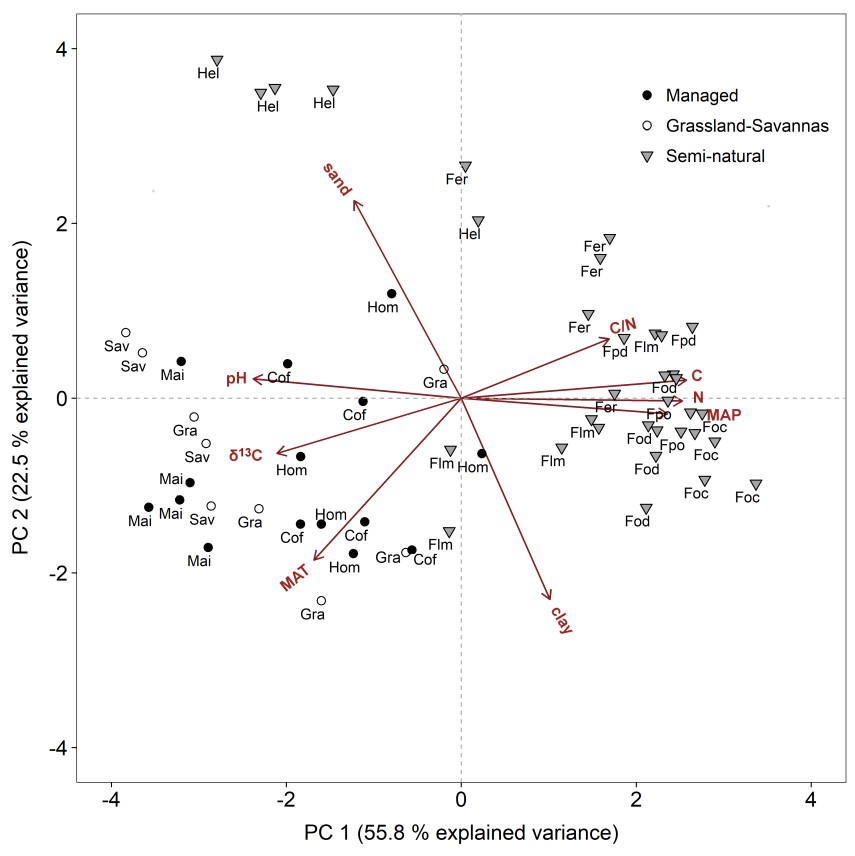

715

**Figure 5:** Principal component analysis bi-plot for soil and climate variables potentially controlling soil $\delta^{15}$N. Symbols are as per all previous figures. Acronyms are as per Table 1. C/N = soil C/N ratio, C = soil carbon content, N = soil nitrogen content, MAP = mean annual precipitation, clay = soil clay content, MAT = mean annual temperature, $\delta^{13}$C = soil $\delta^{13}$C, and pH = soil pH.

720

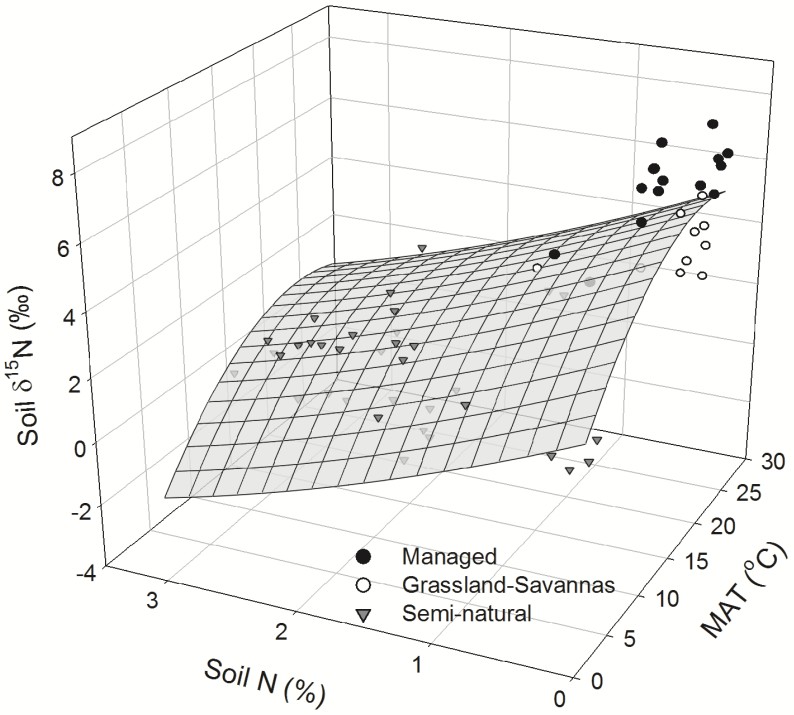

721

**Figure 6:** Measured and modelled soil $\delta^{15}$N values predicted as a function of soil N abundance and mean annual temperature

(MAT). Data points are classified by generic land uses (i.e. intensively managed cropping sites, extensively managed

grassland and savannas, and semi-natural ecosystems) observed along the elevational and land use gradient. The regression

takes the following form: soil $\delta^{15}$N = 1.10 + 0.49 (MAT) − 1.86 (soil N) − 0.01 (MAT)$^2$ + 0.14 (soil N)$^2$; (r² adj= 0.68, P <

0.05, n = 60).