# Peer review of "Stable carbon and nitrogen isotopic composition of leaves, litter, and soils of various ecosystems along an elevational and land-use gradient at Mount Kilimanjaro, Tanzania"

_Biogeosciences, 2018_

## Referee Comment (RC1) · Anonymous Referee #3 · 1 Nov 2018

Review of manuscript bg_2018_407: "Stable carbon and nitrogen isotopic composition of leaves, litter, and soils of various tropical ecosystems along an elevation and land-use gradient at Mount Kilimanjaro, Tanzania" by Gerschlauer et al.

This paper describes the isotopical signature of soils and above ground material in 12 ecosystems at Kilimanjaro. The data obtained is based on a comprehensive sample collection and thus hold a great potential in describing isotopical differences among the

ecosystems. And as an isotopical description of the ecosystems the study surely has fine value, but in order to draw some of the conclusions in the paper, my view is that additional data are needed to fully support those statements. In the general comments below I have tried to suggest some additional data, which the authors ought to include strengthening the paper. I advise the editor to ask the authors for a major revision of the manuscript.

General comments: (1) The authors have a strong focus on using differences in 13C and 15N natural abundance to explain how the different ecosystems work. I really lack some information or estimates of biomass production and balances (both C and N) for the ecosystems. Both for C and N, the input and output of matter would have strong effects on the cycling of those elements, and thus this information is needed to understand/justify the conclusions of the paper.

For example, the authors talk about "tight N cycles" for some ecosystem, but 15N natural abundance cannot stand alone to justify such statement. There we need to include both N inputs and input form, and N removals. It is for example well known that animal manure would affect the 15N natural abundance of soil, and thus, if some of the present ecosystems have grazing animals or animal manure is used e.g. in the homegarden, then this would most likely affect the N signature of the soil. Likewise, for C, we would need to know the annual biomass production to really understand the different 13C natural abundances.

Therefore I ask that the authors in the revised manuscript give actual number or estimates of C and N input and output balances, specify any N fertilizer additions, and make use of this information to support the differences in isotopic signatures.

(2) In the abstract the authors end with a statement regarding "rising temperatures in a changing climate". When I read the manuscript "rising temperatures in a changing climate" is not really clear from the text – please help the reader to understand how this study can say something about "rising temperatures" – many of you ecosystems differ

not only in temperature due to the elevation gradient, but also to e.g. management. Thus, I find it hard to directly understand how "rising temperatures" are covered, unless you can specify that the same ecosystem with similar management is studied at two or more points at the elevation gradient.

In fact, please thoroughly consider your statements regarding "temperature". For example in line 357 you state that "we suggest that . . .increasing temperatures in a changing climate may promote C and N losses" – come on folks isn't that common text book knowledge?

(3) The data from the 12 ecosystems are clustering with the six forest together and the other six ecosystems differing from them. I don't think that all of the statements and comparisons across such clustered data are fair. For example the 13C and 15N natural abundance in forest ecosystems are very alike in spite of quite different temperatures, precipitation, soil C and N contents (Fig. S2 and S3). This to me is interesting – why are they so similar in signature in spite of these differences?

I ask that the authors are more cautious in the data interpretation with such clustered data – as in please don't try to make "correlations across ecosystems", and put some words on where ecosystems have similar isotopic fingerprints. (And why do you forget about the C3 – C4 story in your discussion and presentation of the results?).

(4) The "Helichrysum" ecosystem seems to confuse the authors (and thus also the readers of the manuscript). In one place (line 162-163) the sandy nature is used to "unquestionably" explain soil C and N contents, at another place (line 247-249) lignin is the explaining factor, and in the correlation analysis (Fig. 4, Table S2) also temperature is strongly correlated to the cycling of C and N in this ecosystem. This is confusing, and here I further miss that the authors reflect on their studied ecosystems – the "Helichrysum" ecosystem is a sub-alpine system – where I would guess that temperature play a strong role, not only in C and N turnover processes, but also in biomass production. Thus, I ask the authors to be consistent in their explanation – and please give an

estimate of the biomass production in the ecosystems, so that the reader gets a better picture of the production across the ecosystems.

(5) Table 1 give some basic information regarding the ecosystems. Among other the organic C content, which for the forest ecosystems are at 20-40%. This is quite high. Please clearly specify whether you sampled the O-layer or the upper mineral layer of those soils?

Specific comments: - Title: I would say it is not tropical ecosystems all the way up Kilimanjaro, therefore I think you should consider removing "tropical" from the title.

- In 2.1. Study sites, please include information regarding variables that can affect the C and N signatures. That could be input of N via biological N2 fixation or animal manure (or other fertilizer) and it could be C via biomass production. For example, was the agroforestry based on N2-fixing trees? - In 2.2. Sampling and Analyses. Please make a statement on whether root fragments were visible in the sieved soil. And please in the discussion reflect upon whether unrecovered root material could have affected the soil isotopic signatures (e.g. by using the enrichment of leaves as a proxy for the enrichment of unrecovered roots). - Line 218-219. Please remove this sentence – it is not justified by the figure – there is too much clustering. - Figures and Table: Please keep the same order of the ecosystems all through, and if possible please add the abbreviations for the ecosystems to the legend inside the figure in Figure 1. Also please consider identifying the C3 and C4 dominate ecosystem when presenting 13C natural abundance data. - Figure 5: I don't think I understand what I can learn from this figure. Please explain better or delete it.

―――――――――――――――――――――

---

## Referee Comment (RC2) · Anonymous Referee #4 · 2 Nov 2018

The authors infer nitrogen and carbon cycling dynamics from the nitrogen and carbon stable isotopes of soil and plant samples along an elevational gradient. The gradient in the Mt Kilimanjaro area has a number of variables, including water availability, plant type (C3 and C4) and changes to soils. There are also differences referred to as "ecosystems", where the authors divide the altitudinal gradient into areas as disparate as a 'maize field' versus relatively undisturbed forests. The authors classify these ecosystems and have sufficient samples to examine relationships. The spatial

scale of the study is admirable.

While there is much data here to examine relationships between habitat features and C and N stable isotopes, the relations are correlative. They also rely on inferring what is likely a dynamic process with underlying fluxes from static data. What the authors are relying on is that the isotopes integrate the processes with integrity.

There were several instances where I was concerned about the assumptions and the links the authors were making. First, fertilizers and pesticides could change the d15N, leading to the wrong interpretation of d15 N differences across ecosystems. Is there anything known about this potential artefact? Statements that then follow these N analyses such as "N cycles are tighter" (e.g. L 354) seem too strong. Second, the a priori expectations for d13C patterns was also unclear to me. The paragraph starting L45 was confusing. C3 plants have lighter d13C values but water stress increases the value? How do we think these differences are integrated in Figure 2.

I don't have much in the way of minor edits, etc because I think these broader issues need to be addressed first.

---

## Referee Comment (RC3) · Anonymous Referee #1 · 11 Nov 2018

The authors have develped a good work about nitrogen and carbon cycling dynamics from the nitrogen and carbon stable isotopes of soil and plant samples along an elevational gradient. Due to the remote African's sites where the work has been carried out the data arise in a very important issue about limitation of N availability in ecosystems C sequestration. Methodologically the work is well developed and results a discussion have a good structure that facilitates the reading. I think more works are needed on the multifactorial analyses that implyies soil data, climatological data, and nitrogen and carbon stable isotopes of soil and plants. I not totally sure about authors consideration of grasslands and savannas extensivelly managed and semi-natural ecosystems. I think a little bit information about this clasification would be added. However, authors have been there on field seeing the conditions. As a personal preference, I would like that sites on Lines 162, 166, would be changed by soils. Finally, few minor typographics mistakes would be pointed out: Line 96 –> Kilimanjaro doesn't have capital letter.

———————————————————

---

## Author Comment (AC1) · 25 Dec 2018

Review of manuscript bg_2018_407: "Stable carbon and nitrogen isotopic composition of leaves, litter, and soils of various tropical ecosystems along an elevation and land- use gradient at Mount Kilimanjaro, Tanzania" by Gerschlauer et al.

This paper describes the isotopical signature of soils and above ground material in 12 ecosystems at Kilimanjaro. The data obtained is based on a comprehensive sample collection and thus hold a great potential in describing isotopical differences among the ecosystems. And as an isotopical description of the ecosystems the study surely has fine value, but in order to draw some of the conclusions in the paper, my view is that additional data are needed to fully support those statements. In the general comments below I have tried to suggest some additional data, which the authors ought to include strengthening the paper. I advise the editor to ask the authors for a major revision of the manuscript.

**We thank the reviewer for her/his comments and value their constructive nature. Our answers and comments are in bold font below.**

General comments:

(1) The authors have a strong focus on using differences in 13C and 15N natural abundance to explain how the different ecosystems work. I really lack some information or estimates of biomass production and balances (both C and N) for the ecosystems. Both for C and N, the input and output of matter would have strong effects on the cycling of those elements, and thus this information is needed to understand/justify the conclusions of the paper.

For example, the authors talk about "tight N cycles" for some ecosystem, but 15N natural abundance cannot stand alone to justify such statement. There we need to include both N inputs and input form, and N removals. It is for example well known that animal manure would affect the 15N natural abundance of soil, and thus, if some of the present ecosystems have grazing animals or animal manure is used e.g. in the homegarden, then this would most likely affect the N signature of the soil. Likewise, for C, we would need to know the annual biomass production to really understand the different 13C natural abundances.

Therefore I ask that the authors in the revised manuscript give actual number or estimates of C and N input and output balances, specify any N fertilizer additions, and make use of this information to support the differences in isotopic signatures.

**- We have followed the reviewer's advice and have done our best to provide estimates for biomass production and decomposition rates for all the studied sites.**

**We have made use of relevant research that has been recently published and that assesses plant material decomposition using tea litterbags along the same elevational and land-use gradient (Becker and Kuzyakov, 2018). While we have used the normalized difference vegetation index (NDVI) calculated for these very sites by Röder et al. (2017) as a proxy for primary productivity (Kerr and Ostrovsky, 2003). These indexes provide relevant information on potential ecosystem productivity and decomposition, and are now shown in the new Fig. S1. While there are some estimates of aboveground litterfall for some of these ecosystems (Becker et al., 2015), there is an**

obvious lack of belowground OM inputs, which is a highly significant aspect since they can be up to an order of magnitude larger than aboveground ones. The discussion below has been integrated within the body of the MS, and we include it here for completeness.

Both primary productivity and litter decomposition show a hump-shaped pattern with elevation that resembles that of precipitation. It is interesting to see the close match between the two variables along the elevation range, albeit this trend weakens slightly towards higher elevation sites. Optimum growth and decomposition conditions are shown between 1,800 and 2,500 m.a.s.l.. These locations correspond to low altitude forest ecosystems (Flm and Foc) that do not experience severe seasonal limitations in moisture or temperature as it is otherwise the case in lower as well as higher elevation systems that are moisture and temperature limited respectively (Becker and Kuzyakov, 2018).

It seems reasonable to assume that in the case of natural ecosystems there may be a steady state between SOM inputs and decomposition rates. This should be in contrast with the typically altered nutrient dynamics of disturbed systems, particularly those under agricultural management (Wang et al., 2018). We hypothesized that if carbon inputs and outputs were roughly in balance, then the difference in $\delta^{13}C$ values between plant material and topsoil would be smaller in undisturbed sites compared to managed or disturbed sites. Low fractionation factors in $\delta^{13}C$ are commonly reported between plant material and topsoils in natural systems mainly because of the relatively limited humification of recent organic matter prevalent in topsoils (Acton et al., 2013; Wang et al., 2018). The new Fig. 3 shows relatively small variations in $\delta^{13}C$ enrichment factors (> -1.25 ‰) both in undisturbed semi-natural and extensively managed sites along the elevational gradient, while managed and disturbed sites show higher and more variable $\delta^{13}C$ enrichment factors.

Elevation has a strong influence on the seasonal litterfall dynamics observed in Mt Kilimanjaro, and thus may have significant implications in the SOM cycling across the various ecosystems (Becker et al., 2015). These authors suggest that the large accumulation of particulate organic matter observed at the end of the dry season in low and mid altitude ecosystems may result in the increased mineralization of easily available substrates (Mganga and Kuzyakov, 2014) and nutrient leaching (Gütlein et al., 2018) during the wet season. Therefore, besides the systematic removal of plant biomass characteristic of agricultural systems, annual litterfall patterns may also explain the comparatively lower contents of C and N observed in the topsoils of these managed sites (Table 1). Furthermore, the relationship between $\delta^{13}C$ enrichment factors and soil C/N ratios shown in Fig. 3 may also be quite informative regarding SOM dynamics. As previously mentioned, soil C/N ratios provide a good indication of SOM decomposition processes, typically showing comparatively low values in managed and disturbed systems. These correspond well with sites having large enrichment factors (< -1.25 ‰; i.e. intensively managed and disturbed sites), which agree with the notion of altered SOM dynamics.

- We have also sought the best available information on fertilizer and pesticide use on those sites. We have now included information about the use and isotopic composition of fertilizer and pesticides in a dedicated section in the Supplementary Information.

We would like to acknowledge that contrary to agricultural research stations or purposely-established agricultural field trials, it is extremely difficult to provide reliable estimates of both fertilizers and pesticide rates used in small household farms in sub-Saharan Africa. This is because the actual use of these products is strongly dependent on both its availability in the local/regional market, the economic circumstances of each individual farmer, and individual perceptions about their use (Saiz and Albrecht, 2016). Indeed, a recent study specifically investigating the effect of land use on soil biochemical properties on nearby/comparable sites (Mganga et al., 2016) had to refer to coarse regional estimates of fertilization rates published two decades ago (Giller et al., 1998). Other relevant studies (e.g. Classen et al., 2015; Becker and Kuzyakov, 2018) refer to qualitative estimates compiled by a plant ecologist with long expertise in the region, but no actual amounts of fertilizers or pesticides are provided.

Being well aware of the difficulty to provide accurate numbers on mineral fertilizer and pesticide inputs, we have clearly tagged in the text those sites that receive any of those. These are the two intensively managed systems: Maize (Mai) fields and Coffee (Cof) plantations, and to a lesser extent the homegardens (Hom) sites. In the latter sites Gütlein et al. (2018) report that weed control is mainly done by hand, and the use of mineral or organic N-fertilizers is low or non-existent.

As mentioned earlier, Giller et al. (1998) reported an estimate of ca. 40 kg N ha− inorganic fertilizer use in the Kilimanjaro region. A more recent report (i.e. Senkoro et al., 2017) indicate a generic fertilizer use of 17 kg/ha/yr on a country basis, with about 12% of the national fertilizer share being used in the Kilimanjaro and Arusha regions. Urea (48% N) and diammonium phosphate (18% N) accounted for about half the total volume of fertilizer used in 2010. Nonetheless, the nitrogen isotopic signal of both fertilizers is ~0 ‰ (Bateman and Kelly, 2007), for which it will not provide a significant additional bias on the interpretation of soil $\delta^{15}N$ values. However, the addition of manure ($\delta^{15}N$ ~8 ‰) in Hom systems, albeit used in low quantities (Gütlein et al., 2018), may have well contributed to the high $\delta^{15}N$ values observed in this ecosystem (Fig. 4).

While reliable data on pesticide amounts cannot be provided, we show an indication of two of the most commonly used pesticides as this may serve as a ready reference in future studies. The actual value may strongly depend on the manufacturer, which as in the case of $\delta^{13}C$ can be quite different for glyphosate. Regardless of this, we suggest that the use of pesticides may not pose a strong bias in our isotopic results since their use is limited to intensively managed sites, and the actual isotopic values of pesticides work in the opposite direction to our data (Fig. 4a).

| | $\delta^{13}C$ (‰) | $\delta^{15}N$ (‰) |
|---|---|---|
| Glyphosate | -24 ; -34 [1] | -3.6 [2] |
| Atrazine | -28.9 ; -27.9 [3] | -0.2 ; -1.5 [3] |

[1] Kujawinski, D. M., Wolbert, J. B., Zhang, L., Jochmann, M. A., Widory, D., Baran, N., & Schmidt, T. C. (2013). Carbon isotope ratio measurements of glyphosate and AMPA by liquid chromatography coupled to isotope ratio mass spectrometry. *Analytical and bioanalytical chemistry*, *405*(9), 2869-2878.

[2] Tavares, C. R. D. O., Bendassolli, J. A., Ribeiro, D. N., & Rossete, A. L. R. M. (2010). 15N-labeled glyphosate synthesis and its practical effectiveness. *Scientia Agricola*, *67*(1), 96-101

[3] Meyer, A. H., Penning, H., Lowag, H., & Elsner, M. (2008). Precise and accurate compound specific carbon and nitrogen isotope analysis of atrazine: critical role of combustion oven conditions. *Environmental science & technology*, *42*(21), 7757-7763.

- We have also included information on other ecosystem inputs in our response to a dedicated specific comment about section 2.1.

We trust that the reader has now sufficient information to critically assess the limitations that the study contains on external nutrient additions.

*References:*

Acton, P., Fox, J., Campbell, E., Rowe, H., & Wilkinson, M. (2013). Carbon isotopes for estimating soil decomposition and physical mixing in well-drained forest soils. *Journal of Geophysical Research: Biogeosciences*, *118*(4), 1532-1545.

Bateman, A. S., and Kelly, S. D. (2007). Fertilizer nitrogen isotope signatures. *Isotopes in environmental and health studies*, *43*(3), 237-247.

Becker, J., Pabst, H., Mnyonga, J., and Kuzyakov, Y. (2015). Annual litter fall dynamics and nutrient deposition depending on elevation and land use at Mt. Kilimanjaro. *Biogeosciences*, 12, 5635–5646

Becker, J. N., and Kuzyakov, Y. (2018). Teatime on Mount Kilimanjaro: Assessing climate and land-use effects on litter decomposition and stabilization using the Tea Bag Index. *Land Degradation & Development*, *29*(8), 2321-2329

Classen et al. (2015). Temperature versus resource constraints: Which factors determine bee diversity on Mount Kilimanjaro, Tanzania? Global Ecology and Biogeography, 24, 642–652.

Kerr, J. T., and Ostrovsky, M. (2003). From space to species: ecological applications for remote sensing. *Trends in ecology & evolution*, *18*(6), 299-305.

Giller et al. (1998). Environmental constraints to nodulation and nitrogen fixation of Phaseolus vulgaris L in Tanzania II. Response to N and P fertilizers and inoculation with Rhizobium. *African Crop Science Journal*, *6*(2), 171-178.

Gütlein et al (2018). Impacts of climate and land use on N2O and CH4 fluxes from tropical ecosystems in the Mt. Kilimanjaro region, Tanzania. Glob. Change Biol. 24, 1239–1255.

Mganga, K. Z., Razavi, B. S., and Kuzyakov, Y. (2016). Land use affects soil biochemical properties in Mt. Kilimanjaro region. *Catena*, *141*, 22-29.

Mganga, K. Z., and Kuzyakov, Y. (2014). Glucose decomposition and its incorporation into soil microbial biomass depending on land use in Mt. Kilimanjaro ecosystems. *European Journal of Soil Biology*, 62, 74–82

Röder, J., Detsch, F., Otte, I., Appelhans, T., Nauss, T., Peters, M. K., & Brandl, R. (2017). Heterogeneous patterns of abundance of epigeic arthropod taxa along a major elevation gradient. *Biotropica*, *49*(2), 217-228.

Saiz, G., and Albrecht, A. (2016). Methods for smallholder quantification of soil carbon stocks and stock changes. *In:* Rosenstock TS, Rufino MC, Butterbach-Bahl K, Wollenberg E, Richards M (eds) *Measurement methods Standard Assessment Of Agricultural Mitigation Potential And Livelihoods (SAMPLES)*. ISBN 978-3-319-29792-7. CGIAR Research Program on Climate Change, Agriculture and Food Security. pp 135-162.

Senkoro et al (2017). Optimizing fertilizer use within the context of integrated soil fertility management in Tanzania. *Fertilizer use optimization in Sub-Saharan Africa. CAB International, Nairobi, Kenya*, 176-192.

**Wang, C., Houlton, B. Z., Liu, D., Hou, J., Cheng, W., & Bai, E. (2018). Stable isotopic constraints on global soil organic carbon turnover.** *Biogeosciences*, *15*(4), 987-995

(2) In the abstract the authors end with a statement regarding "rising temperatures in a changing climate". When I read the manuscript "rising temperatures in a changing climate" is not really clear from the text – please help the reader to understand how this study can say something about "rising temperatures" – many of you ecosystems differ not only in temperature due to the elevation gradient, but also to e.g. management. Thus, I find it hard to directly understand how "rising temperatures" are covered, unless you can specify that the same ecosystem with similar management is studied at two or more points at the elevation gradient.

In fact, please thoroughly consider your statements regarding "temperature". For example in line 357 you state that "we suggest that . . .increasing temperatures in a changing climate may promote C and N losses" – come on folks isn't that common text book knowledge?

**As mentioned by the reviewer, our study does not specifically assess the effect of rising temperatures on SOM dynamics. However, our data show strong relationships between temperature and variables directly related to SOM dynamics such as soil $\delta^{13}C$, C, N and C/N ratios. These results agree well with recent findings by Becker and Kuzyakov (2018) who studied SOM decomposition dynamics at these very sites. An important finding revealed by that study is that of seasonal variation in temperature is a major controlling factor in litter decomposition. Their study shows that small seasonal variations in temperature observed at high elevation sites exert a strong effect on litter decomposition rates. Therefore, the authors argue that the projected increase in surface temperature may result in potentially large soil C losses at high elevation sites due to their strong temperature sensitivity to decomposition. This is normally expected since the temperature sensitivity of decomposition is generally higher at higher elevations and at low temperatures (Blagodatskaya et al., 2016; Davidson and Janssens, 2006).**

**We believe that the data obtained in our study reinforces such view. Please note we use the term 'suggest' to refer to this aspect. In any case, we are ready to remove this statement if the reviewer still has a concern with it.**

**Blagodatskaya, E., Blagodatsky, S., Khomyakov, N., Myachina, O., & Kuzyakov, Y. (2016). Temperature sensitivity and enzymatic mechanisms of soil organic matter decomposition along an altitudinal gradient on Mount Kilimanjaro.** *Scientific Reports*, **6, 22240.**

(3) The data from the 12 ecosystems are clustering with the six forest together and the other six ecosystems differing from them. I don't think that all of the statements and comparisons across such clustered data are fair. For example the 13C and 15N natural abundance in forest ecosystems are very alike in spite of quite different temperatures, precipitation, soil C and N contents (Fig. S2 and S3). This to me is interesting – why are they so similar in signature in spite of these differences?

I ask that the authors are more cautious in the data interpretation with such clustered data – as in please don't try to make "correlations across ecosystems", and put some words on

where ecosystems have similar isotopic fingerprints. (And why do you forget about the C3 – C4 story in your discussion and presentation of the results?).

**Ecosystems dominated by C3 vegetation, such montane tropical forests, usually show a relatively small increase in $\delta^{13}C$ values of about 1.2‰ per 1,000 m elevation (Körner et al., 1988; Bird et al., 1994). Such trend has been graphically depicted in Fig. S2 to allow for direct comparisons with our data. The text in the relevant section (4.1) has been significantly edited to improve the discussion of our results.**

**Connected to our response to the first comment, explaining the estimates of ecosystem productivity and decomposition, the new figure showing the relationship between $\delta^{13}C$ enrichment factors and soil C/N ratios and soil carbon contents (Fig. 3) further support the contrasting SOM dynamics between semi-natural ecosystems and intensively managed/disturbed systems.**

**A final comment on the similar $\delta^{13}C$ values in forest ecosystems: work conducted along a comparable elevation range by Bird et al. (1994) in Papua New Guinea shows a negative relationship between soil $\delta^{13}C$ corrected for altitude and SOC contents in C3-only vegetation systems, which roughly resembles our data and relies on similar explanations. Thus, we were not overly surprised with the relatively small variation in soil $\delta^{13}C$ values and the moderate range in SOC contents observed along the environmental conditions encompassed by these semi-natural forest ecosystems.**

**While it is widely accepted that soil $\delta^{15}N$ provides valuable insights about the N cycle in a given ecosystem, we agree with the reviewer that a number of factors including the nature and balance of N inputs and outputs may significantly affect its isotopic signal, thus rendering it not sufficient to undisputedly draw the conclusion about open and closed nitrogen cycles we had made in section 4.3 (and in the abstract). We do thank the reviewer for having brought this important aspect up. Indeed, after considering the water concentrations of soil nitrate provided by Gütlein et al (2018), it appears that forest ecosystems have significant N losses through this pathway, which would go unnoticed if one relies exclusively on soil $\delta^{15}N$ values as was the case in the study by Zech et al (2011). Consequently, we have modified our statements regarding the open and close N cycles in the abstract, discussion and the conclusions.**

(4) The "Helichrysum" ecosystem seems to confuse the authors (and thus also the readers of the manuscript). In one place (line 162-163) the sandy nature is used to "unquestionably" explain soil C and N contents, at another place (line 247-249) lignin is the explaining factor, and in the correlation analysis (Fig. 4, Table S2) also temperature is strongly correlated to the cycling of C and N in this ecosystem. This is confusing, and here I further miss that the authors reflect on their studied ecosystems – the "Helichrysum" ecosystem is a sub-alpine system – where I would guess that temperature play a strong role, not only in C and N turnover processes, but also in biomass production. Thus, I ask the authors to be consistent in their explanation – and please give an estimate of the biomass production in the ecosystems, so that the reader gets a better picture of the production across the ecosystems.

**Fig S1 shows that the *Helichrysum* is the only ecosystem where decomposition potential is higher than production. We agree with the reviewer that the limited productivity shown by this ecosystem is strongly influenced by its low temperature. We have amended the text describing general soil characteristics to incorporate such**

**fact and now reads: "The low temperatures and sandy nature of the *Helichrysum* sites play a strong role in their characteristically low productivity and moderate decomposition potentials (Table 1; Fig. S1), which unquestionably affects the comparatively low soil C and N contents of these alpine systems'.**

**The above discussion is specifically about soil C and N contents in *Helichrysum* sites. However, the lignin explanation focuses on $\delta^{13}$C values and connects to the previous point (3) raised by the reviewer. The MS text reads: "Further variations in soil $\delta^{13}$C values could also be related to the biochemical composition of the precursor biomass. For instance, herbaceous vegetation is pervasive at high elevations, and contains relatively low amounts of lignin – an organic compound characteristically depleted in $^{13}$C (Benner et al., 1987). This may contribute to explain the higher $\delta^{13}$C values observed in plant and soil materials in alpine ecosystems dominated by *Helichrysum* vegetation, compared to forest ecosystems at lower elevations (Fig. 2)".**

(5) Table 1 give some basic information regarding the ecosystems. Among other the organic C content, which for the forest ecosystems are at 20-40%. This is quite high. Please clearly specify whether you sampled the O-layer or the upper mineral layer of those soils?

**We sampled the upper mineral layer of the soils.**

Specific comments:

- Title: I would say it is not tropical ecosystems all the way up Kilimanjaro, therefore I think you should consider removing "tropical" from the title.

**Revised as suggested.**

- In 2.1. Study sites, please include information regarding variables that can affect the C and N signatures. That could be input of N via biological N2 fixation or animal manure (or other fertilizer) and it could be C via biomass production. For example, was the agroforestry based on N2-fixing trees?

**Connected to our response to the first comment, and notwithstanding the obvious practical limitations of a study of such scope and nature, we have now included relevant information on potential ecosystem productivity and decomposition. Moreover, admitting the challenge in providing accurate numbers on mineral fertilizer and pesticide inputs, we have clearly tagged in the text those sites that receive any. These are the two monocultures: Maize (Mai) fields and Coffee (Cof) plantations, and to a lesser extent the homegardens (Hom) sites. Extensively managed sites (i.e. Sav and Gra) receive varying amounts of organic inputs from grazing animals, but again, the actual rates are unknown.**

**The traditional agroforestry systems (Hom) maintain a forest-like structure consisting of indigenous forest species that includes *Albizia schimperi*, a tree that may potentially fix atmospheric N. This is one of the 5 most abundant species in 2 and 4 of the Hom and Cof sites respectively, making up less than 25% of the vegetation cover in all cases.**

- In 2.2. Sampling and Analyses. Please make a statement on whether root fragments were visible in the sieved soil. And please in the discussion reflect upon whether unrecovered

root material could have affected the soil isotopic signatures (e.g. by using the enrichment of leaves as a proxy for the enrichment of unrecovered roots).

**We have added a specific statement in M&M that reads: "Soil was sieved to 2 mm with visible root fragments being further removed prior to grinding". Furthermore, following the reviewer's advice we have estimated the effect that the removal of visible sieved roots might have caused on soil isotopic values. We re-calculated soil isotope values by mass balance making the following assumptions. In addition to taking leaf isotope values as a proxy for roots as suggested by the reviewer, a non-conservative assumption was made about average root mass (< 2 mm) being ~5% of the total mass in the sample (w/w). This is above double the maximum value observed by Saiz et al (2012) for roots > 2mm contained in soil samples collected from contrasting tropical ecosystems.**

**Re-calculated soil $\delta^{13}C$ and $\delta^{15}N$ values under the assumptions referred above were on average 0.15 and 0.17‰ higher than the original soil isotopic values, which are even lower than the analytical error (0.2 ‰). We have added a specific mention to this in the discussion.**

**Saiz, G., Bird, M., Domingues, T., Schrodt, F., Schwarz, M., Feldpausch, T., Veenendaal, E., Djagbletey, G., Hien, F., Compaore, H., Diallo, A., Lloyd, J.: Variation in soil carbon stocks and their determinants across a precipitation gradient in West Africa. Global Change Biology 18, 1670-1683. doi:10.1111/j.1365-2486.2012.02657.x, 2012.**

- Line 218-219. Please remove this sentence – it is not justified by the figure – there is too much clustering.

**We have deleted this sentence.**

- Figures and Table: Please keep the same order of the ecosystems all through, and if possible please add the abbreviations for the ecosystems to the legend inside the figure in Figure 1. Also please consider identifying the C3 and C4 dominate ecosystem when presenting 13C natural abundance data.

**We have revised the order of appearance of sites. We have modified Table 1 making sure that all sites appear in the same order both in Figures and Tables. We have we have also included sites' abbreviations in Fig. 1 legend.**

- Figure 5: I don't think I understand what I can learn from this figure. Please explain better or delete it.

**We have now improved the discussion on this figure (now Fig. 6) in section 4.3.**

---

## Author Comment (AC2) · 26 Dec 2018

The authors infer nitrogen and carbon cycling dynamics from the nitrogen and carbon stable isotopes of soil and plant samples along an elevational gradient. The gradient in the Mt Kilimanjaro area has a number of variables, including water availability, plant type (C3 and C4) and changes to soils. There are also differences referred to as "ecosystems", where the authors divide the altitudinal gradient into areas as disparate as a 'maize field' versus relatively undisturbed forests. The authors classify these ecosystems and have sufficient samples to examine relationships. The spatial scale of the study is admirable.

While there is much data here to examine relationships between habitat features and C and N stable isotopes, the relations are correlative. They also rely on inferring what is likely a dynamic process with underlying fluxes from static data. What the authors are relying on is that the isotopes integrate the processes with integrity.

**We thank the reviewer for her/his positive comments. We also appreciate the criticism, which we address in bold font below.**

There were several instances where I was concerned about the assumptions and the links the authors were making. First, fertilizers and pesticides could change the d15N, leading to the wrong interpretation of d15 N differences across ecosystems. Is there anything known about this potential artefact? Statements that then follow these N analyses such as "N cycles are tighter" (e.g. L 354) seem too strong.

**We agree that the use of fertilizer and pesticides may pose a bias on the results and their subsequent interpretation. As explained in our answers to reviewer #3, we have clearly tagged and discussed those sites that have had external applications of fertilizers (both organic and mineral) as well as pesticides. We have also included information about the use and isotopic composition of fertilizer and pesticides in a dedicated section in the Supplementary Information, and included information on N-fixing trees.**

**We trust that the reader has now sufficient information to critically assess the limitations that the study contains on external nutrient additions.**

**The discussion on the N cycle as supported by soil $\delta^{15}$N values, was also a criticism shared by reviewer #3. We also thank this reviewer for having raised this important aspect. Indeed, after considering the water concentrations of soil nitrate provided by Gütlein et al (2018), it appears that forest ecosystems have significant N losses through this pathway, which would go unnoticed if one relies exclusively on soil $\delta^{15}$N values as was the case in the study by Zech et al (2011). Consequently, we have modified our statements regarding the open and close N cycles in the abstract, discussion and the conclusions.**

Second, the a priori expectations for d13C patterns was also unclear to me. The paragraph starting L45 was confusing. C3 plants have lighter d13C values but water stress increases the value? How do we think these differences are integrated in Figure 2.

I don't have much in the way of minor edits, etc because I think these broader issues need to be addressed first.

**The paragraph starting in Line 45 is a general introduction about the variation of $\delta^{13}C$ values on plants. In the referred paragraph we do state that C3 plants do show lighter $\delta^{13}C$ values than their C4 counterparts. The relative abundance of C3 and C4 plants greatly determines the $\delta^{13}C$ of a given ecosystem, which greatly explains the large variation exhibited by managed sites with mixed C3/C4 vegetation located at lower elevations.**

**Our sites have been categorized according to land use intensities (i.e. managed and semi-natural) following a similar classification used by Classen et al. (2015) and Schellenberger Costa et al. (2017), which employed factors as land use, vegetation structure, annual biomass removal, input of fertilizers and pesticides.**

**We see pertinent to reiterate (as it has been explained in the MS text), that all semi-natural sites are $C_3$-dominated ecosystems. If one just considers those ecosystems (nearly or exclusively) composed by C3 plants ($\delta^{13}C$ values <-24 ‰ ~ –semi-natural ecosystems occurring above 1,800 m.a.s.l.), the effect of increasing $\delta^{13}C$ values with altitude is quite noticeable (Fig. S2), and corresponds with a decreasing trend in MAP (Fig. S3 b). Fig. 2 shows the variation in $\delta^{13}C$ values of plants, litter and soil samples along the elevational and land use gradient. As such, the figure does not directly show the variation in $\delta^{13}C$ values with precipitation. Rather, this is shown in Fig. S3 b.**

**Finally, we would also like to state that it is abundantly clear that water deficits may cause the enrichment of $^{13}C$ in $C_3$ plants (Farquhar and Sharkey, 1982; Kohn, 2010; Körner et al., 1991). Therefore, we do not see any discrepancy with the referred introductory statement and our results.**

**Note: The MS text (and to a lesser extent Fig. 1) explain the distribution of precipitation along the elevation gradient "Maximum mean annual precipitation (MAP) of 2,552 mm occurs at an elevation of around 2,260 m a.s.l., decreasing towards lower as well as higher elevations, reaching 657 and 1,208 mm $y^{-1}$ at 871 and 4,550 m respectively (Table 1)".**

*References:*

**Classen, A., Peters, M. K., Kindeketa, W. J., Appelhans, T., Eardley, C. D., Gikungu, M. W., ... & Steffan-Dewenter, I. (2015). Temperature versus resource constraints: which factors determine bee diversity on M ount K ilimanjaro, T anzania?. *Global Ecology and Biogeography*, *24*(6), 642-652.**

**Farquhar, G.D., Sharkey, T.D.: Stomatal Conductance and Photosynthesis. Annu. Rev. Plant Physiol. 33, 317–345. doi.org/10.1146/annurev.pp.33.060182.001533, 1982.**

**Kohn, M.J. Carbon isotope compositions of terrestrial C3 plants as indicators of (paleo)ecology and (paleo)climate. Proc. Natl. Acad. Sci. 107, 19691–19695. doi.org/10.1073/pnas.1004933107, 2010.**

**Körner, C., Farquhar, G.D., Wong, S.C.: Carbon Isotope Discrimination by Plants Follows Latitudinal and Altitudinal Trends. Oecologia 88, 30–40 , 1991.**

**Schellenberger Costa, D., Gerschlauer, F., Pabst, H., Kühnel, A., Huwe, B., Kiese, R., Kuzyakov, Y., Kleyer, M. and Kühn, I.: Community-weighted means and functional dispersion of plant functional traits along environmental gradients on Mount Kilimanjaro, J. Veg. Sci., 28(4), 684–695, doi:10.1111/jvs.12542, 2017.**

---

## Author Comment (AC3) · 26 Dec 2018

The authors have develped a good work about nitrogen and carbon cycling dynamics from the nitrogen and carbon stable isotopes of soil and plant samples along an elevational gradient. Due to the remote African's sites where the work has been carried out the data arise in a very important issue about limitation of N availability in ecosystems C sequestration. Methodologically the work is well developed and results a discussion have a good structure that facilitates the reading. I think more works are needed on the multifactorial analyses that implyies soil data, climatological data, and nitrogen and carbon stable isotopes of soil and plants.

**We thank the reviewer for her/his positive comments. We provide our answers in bold font below.**

I not totally sure about authors consideration of grasslands and savannas extensivelly managed and semi-natural ecosystems. I think a little bit information about this clasification would be added. However, authors have been there on field seeing the conditions.

**The classification we use has been followed by previous research working on the same sites (e.g. Becker and Kuzyakov, 2018; Classen et al., 2015; Ensslin et al., 2015; Gerschlauer et al., 2016; Gütlein et al., 2018; Mganga et al., 2014), and agree with our observation in the field.**

**These references are on the MS reference list.**

As a personal preference, I would like that sites on Lines 162, 166, would be changed by soils.

**We replaced sites with soils as suggested, but we felt that the change worsened the reading of the sentences.**

Finally, few minor typographics mistakes would be pointed out: Line 96 –> Kilimanjaro doesn't have capital letter.

**Revised as suggested**

---

## Author Comment (AC4) · 26 Dec 2018

[revised manuscript text omitted]

**Appendix – Fertilizer and pesticide isotopic composition**

*Fertilizers*

A general indication of fertilizer used in the region is provided here.

Giller et al. (1998) reported an estimate of ca. 40 kg N ha$^{-1}$ inorganic fertilizer use in the Kilimanjaro region. A more recent report by Senkoro et al. (2017) indicate a generic fertilizer use of 17 kg ha$^{-1}$ y$^{-1}$ on a country basis, with about 12% of the national fertilizer share being used in the Kilimanjaro and Arusha regions. Urea (48% N) and diammonium phosphate (18% N) accounted for about half the total volume of fertilizer used in 2010. The nitrogen isotopic values of both fertilizers is ~0 ‰ (Bateman and Kelly, 2007), and as such does not pose a significant additional bias on the interpretation of soil $\delta^{15}$N values. However, the addition of manure ($\delta^{15}$N ~8 ‰) in Hom systems, albeit used in low quantities (Gütlein et al., 2018), may have well contributed to the high $\delta^{15}$N values observed in this ecosystem (Fig. 4).

Bateman, A. S., and Kelly, S. D. (2007). Fertilizer nitrogen isotope signatures. *Isotopes in environmental and health studies*, *43*(3), 237-247.

Giller et al. (1998). Environmental constraints to nodulation and nitrogen fixation of Phaseolus vulgaris L in Tanzania II. Response to N and P fertilizers and inoculation with Rhizobium. *African Crop Science Journal*, *6*(2), 171-178.

Gütlein et al (2018). Impacts of climate and land use on N2O and CH4 fluxes from tropical ecosystems in the Mt. Kilimanjaro region, Tanzania. Glob. Change Biol. 24, 1239–1255.

Senkoro et al (2017). Optimizing fertilizer use within the context of integrated soil fertility management in Tanzania. *Fertilizer use optimization in Sub-Saharan Africa. CAB International, Nairobi, Kenya*, 176-192.

*Pesticides*

The isotopic values of the two most commonly used pesticides are shown below. The actual product values may strongly depend on the manufacturer, which as in the case of $\delta^{13}$C can be quite different for glyphosate.

|  | $\delta^{13}$C (‰) | $\delta^{15}$N (‰) |
|---|---|---|
| Glyphosate | -24.0 ; -34.0 [1] | -3.6 [2] |
| Atrazine | -28.9 ; -27.9 [3] | -0.2 ; -1.5 [3] |

[1] Kujawinski, D. M., Wolbert, J. B., Zhang, L., Jochmann, M. A., Widory, D., Baran, N., & Schmidt, T. C. (2013). Carbon isotope ratio measurements of glyphosate and AMPA by liquid chromatography coupled to isotope ratio mass spectrometry. *Analytical and bioanalytical chemistry*, *405*(9), 2869-2878.

[2] Tavares, C. R. D. O., Bendassolli, J. A., Ribeiro, D. N., & Rossete, A. L. R. M. (2010). 15N-labeled glyphosate synthesis and its practical effectiveness. *Scientia Agricola*, *67*(1), 96-101

[3] Meyer, A. H., Penning, H., Lowag, H., & Elsner, M. (2008). Precise and accurate compound specific carbon and nitrogen isotope analysis of atrazine: critical role of combustion oven conditions. *Environmental science & technology*, *42*(21), 7757-7763.

**Table S1** Pearson's correlations coefficients (r) between soil, litter, leaf, and climatic parameters. Correlation analysis was conducted with all five replicates of each of the twelve ecosystems (n = 60)

| | Variable | Soil δ¹⁵N | Soil N content | Soil δ¹³C | Soil C content | Soil C/N ratio | Litter δ¹⁵N | Litter N content | Litter δ¹³C | Litter C content | Litter C/N ratio | Leaf δ¹⁵N | Leaf N content | Leaf δ¹³C | Leaf C content | Leaf C/N ratio |
|---|---|---|---|---|---|---|---|---|---|---|---|---|---|---|---|---|
| **Soil** | δ¹⁵N | | -0.70*** | 0.52*** | -0.76*** | -0.54*** | 0.82*** | -0.13 | 0.44*** | -0.72*** | -0.06 | 0.75*** | 0.21 | 0.47*** | -0.38** | -0.27* |
| | N content | | | -0.63*** | 0.96*** | 0.38** | -0.44*** | 0.49*** | -0.56*** | 0.72*** | -0.26* | -0.38** | 0.21 | -0.61*** | 0.34** | -0.15 |
| | δ¹³C | | | | -0.61*** | 0.01 | 0.18 | -0.60*** | 0.79*** | -0.43*** | 0.51*** | 0.15 | -0.31* | 0.76*** | -0.49*** | 0.28* |
| | C content | | | | | 0.56*** | -0.53*** | 0.38** | -0.54*** | 0.76*** | -0.17 | -0.45*** | 0.07 | -0.59*** | 0.42*** | -0.04 |
| | C/N ratio | | | | | | -0.59*** | -0.19 | -0.15 | 0.51*** | 0.303* | -0.54*** | -0.40** | -0.14 | 0.39** | 0.40** |
| **Litter** | δ¹⁵N | | | | | | | 0.26* | 0.13 | -0.68*** | -0.48*** | 0.92*** | 0.53*** | 0.20 | -0.25 | -0.57*** |
| | N content | | | | | | | | -0.66*** | 0.26* | -0.87*** | 0.26* | 0.73*** | -0.61*** | 0.21 | -0.64*** |
| | δ¹³C | | | | | | | | | -0.42*** | 0.54*** | 0.14 | -0.36** | 0.88*** | -0.54*** | 0.22 |
| | C content | | | | | | | | | | 0.11 | -0.57*** | -0.05 | -0.49*** | 0.39** | 0.08 |
| | C/N ratio | | | | | | | | | | | -0.42*** | -0.69*** | 0.47*** | -0.12 | 0.63*** |
| **Leaf** | δ¹⁵N | | | | | | | | | | | | 0.53*** | 0.17 | -0.17 | -0.61*** |
| | N content | | | | | | | | | | | | | -0.44*** | -0.13 | -0.92*** |
| | δ¹³C | | | | | | | | | | | | | | -0.44*** | 0.30* |
| | C content | | | | | | | | | | | | | | | 0.19 |
| | C/N ratio | | | | | | | | | | | | | | | |
| **Soil** | pH | 0.51*** | -0.76*** | 0.65*** | -0.78*** | -0.28* | 0.26* | -0.51*** | 0.44*** | -0.55*** | 0.34** | 0.20 | -0.24 | 0.45*** | -0.40** | 0.26* |
| | clay content | 0.14 | 0.33** | -0.23 | 0.27* | -0.10 | 0.32* | 0.37** | -0.12 | 0.02 | -0.34** | 0.31* | 0.44*** | -0.16 | -0.06 | -0.46*** |
| | silt content | 0.01 | 0.27* | -0.04 | 0.30* | 0.20 | 0.08 | 0.22 | 0.02 | 0.14 | -0.23 | 0.09 | 0.15 | -0.01 | 0.05 | -0.24 |
| | sand content | -0.12 | -0.43*** | 0.22 | -0.39** | -0.04 | -0.31* | -0.43*** | 0.09 | -0.10 | 0.41** | -0.31* | -0.45*** | 0.14 | 0.02 | 0.52*** |
| **MAP** | | -0.60*** | 0.81*** | -0.72*** | 0.76*** | 0.19 | -0.32* | 0.58*** | -0.65*** | 0.50*** | -0.44*** | -0.27* | 0.33** | -0.60*** | 0.34** | -0.26* |
| **MAT** | | 0.73*** | -0.54*** | 0.66*** | -0.60*** | -0.33** | 0.67*** | -0.16 | 0.55*** | -0.62*** | 0.05 | 0.61*** | 0.25 | 0.55*** | -0.48*** | -0.33* |

Levels of significance: * P < 0.05, ** P < 0.01, *** P < 0.001

**Table S2** Correlation coefficients (r) and P values of selected variables included in the principal component analysis used to identify the main factors driving soil $\delta^{15}N$. Only variables showing r > 0.5 were considered

| Principal component | Variable | r | P value |
|---|---|---|---|
| PC 1 | Soil C content | 0.93 | <0.001 |
| | Soil N content | 0.93 | <0.001 |
| | Soil C/N ratio | 0.61 | <0.001 |
| | Soil pH | -0.87 | <0.001 |
| | Soil $\delta^{13}C$ | -0.76 | <0.001 |
| | MAP | 0.87 | <0.001 |
| | MAT | -0.63 | <0.001 |
| PC 2 | Soil clay content | -0.84 | <0.001 |
| | Soil sand content | 0.82 | <0.001 |
| | MAT | -0.65 | <0.001 |

[Figure]

**Fig. S1** Annual means of Tea Bag Index decomposition rate constant (k) reported by Becker and Kuzyakov (2018), and Normalized Difference Vegetation Index (NDVI) calculated by Röder et al. (2017) as a proxy for primary productivity (Kerr and Ostrovsky, 2003) for the same ecosystems studied in the Kilimanjaro land-use and elevational gradient. Solid and dotted line corresponds to k and NDVI 3rd degree polynomial regressions; $r^2$ 0.82 and 0.78 respectively.

[Figure]

**Fig. S2** Variation in $\delta^{13}C$ values along the Kilimanjaro land-use and elevational gradient for leaves, litter, and soil. Solid symbols denote semi-natural ecosystems, while open symbols correspond to managed ecosystems. The dotted line represents the theoretical global relationship between altitude and $\delta^{13}C$ of plant leaves ($C_3$ vegetation only) developed by Körner et al. (1988) and is shown here for reference. The ecosystem acronyms used are as per Table 1. Mai, Cof, and Hom are managed cropping sites, Gra and Sav are extensively managed grasslands and savannas, while the rest represent semi-natural ecosystems. *Reference:* Körner, C., Farquhar, G.D., Roksandic, Z., 1988. A global survey of carbon isotope discrimination in plants from high altitude. Oecologia 74, 623–632. https://doi.org/10.1007/BF00380063.

Saiz, Gustavo 25.12.2018 18:34

[Figure]

**Fig. S3** Relationship between soil δ¹³C values and mean annual temperature (a), mean annual precipitation (b), soil organic carbon (c), and soil C/N ratios (d) for all ecosystems. Each data point represents the average of five sites, and bars denote standard error of the means. Symbols are as per all previous figures. The ecosystem acronyms used are as per Table 1.

Saiz, Gustavo 25.12.2018 18:34

[Figure]

**Fig. S4** Relationship between soil $\delta^{15}N$ values and mean annual temperature (a), mean annual precipitation (b), soil nitrogen (c), and soil C/N ratios (d) for all ecosystems. Each data point represents the average of five sites, and bars denote standard error of the means. Symbols are as per all previous figures. The ecosystem acronyms used are as per Table 1.

Saiz, Gustavo 25.12.2018 18:34